# LOCAL-GLOBAL SHORTEST PATH ALGORITHMS ON RANDOM GRAPHS, ENHANCED WITH GNNS

## ABSTRACT

Graph neural networks (GNNs) using local message passing were recently shown to inherit the intrinsic limitations of local algorithms in solving combinatorial graph optimization problems such as finding shortest distances (Loukas, 2020). To address this issue, Awasthi et al. (2022) proposed architectures based on Bourgain's (1985) seminal work on Hilbert space embeddings. These architectures enhance local message passing in GNNs with a single global computation, yielding a local-global algorithm. This paper focuses on the average-case analysis of more general local-global algorithms for finding shortest distances (of which GNN+ is a particular case). Our primary contribution is a theoretical analysis of these algorithms on Erdős-Rényi (ER) random graphs. We prove that, on random graphs, these algorithms have lower distortion of shortest distances for most pairs of nodes w.h.p. while requiring a lower embedding dimension. Inspired by Awasthi et al. (2022), and to automate local computations and improve computational efficiency in practical scenarios, we further propose a modification to these algorithms that incorporates GNNs in the local computation phase. Empirical results on ER graphs and benchmark graph datasets demonstrate the enhanced performance of the GNN-augmented algorithm over the traditional approach.

## 1 INTRODUCTION

Finding shortest paths on networks is an important combinatorial optimization problem arising in many practical applications, such as transportation networks Fu et al. (2006) and integrated circuit design Cong et al. (1998). Unlike other optimization problems on graphs, exact solutions for shortest paths can be found using classical algorithms such as Dijkstra's algorithm in polynomial time. Moreover, advancements in indexing techniques have made exact shortest-path distance queries highly efficient, with solutions capable of handling large-scale graphs and providing microsecond-level query times in certain settings (Akiba et al., 2013; Hayashi et al., 2016; Ouyang et al., 2018; Farhan et al., 2018).

However, not all scenarios allow for such efficient indexing. For example, dynamic networks with frequently updated edge weights or applications requiring real-time computation on resource-constrained devices may not benefit from precomputed indexes. In such cases, approximate methods are particularly valuable due to their adaptability and lower computational overhead. This has motivated the exploration of machine learning approaches to shortest path finding, particularly those employing graph neural networks (GNNs).

Despite their promise in combinatorial optimization (Lemos et al., 2019; Cappart et al., 2023; Li et al., 2018; 2023; Veličković et al., 2019; Zhang et al., 2023), GNN-based approaches face significant challenges in the shortest path problem. Local message-passing algorithms like GNNs are constrained by impossibility results (Loukas, 2020; Sarma et al., 2012), requiring prohibitively large embedding dimensions or numbers of convolutions to achieve even a constant-factor approximation of distances in the worst-case. A promising direction to address these limitations is the combination of local message-passing and global methods, which can provide a better tradeoff between efficiency and accuracy. For example, Awasthi et al. (2022) propose GNN+, a two-part architecture where GNNs compute local path distances, and a global fully connected layer combines their outputs (Awasthi et al., 2022). Fundamentally, such approaches are inspired by Bourgain's seminal result on metric space embeddings into Hilbert spaces (Bourgain, 1985), which quantifies the error incurred when

approximating shortest path distances with sums or differences of local embeddings. Bourgain's theorem also prescribes minimum sketch sizes for the class of local-global algorithms that includes GNN+.

While these algorithms show strong empirical performance, their theoretical underpinnings remain sparse. Existing results, including those of Bourgain (Bourgain, 1985), Matoušek (Matoušek, 1996), and Das Sarma et al. (Sarma et al., 2012), are worst-case guarantees. Empirical evidence suggests these worst-case bounds can be overly pessimistic for typical graphs, highlighting the need for theoretical guarantees tailored to average-case graphs.

**Theoretical contributions.** This paper focuses on the theoretical analysis of local-global algorithms inspired by Bourgain's embedding theorem on Erdös–Rényi (ER) random graphs. ER graphs are a foundational model in the random graphs literature, offering insights into average-case scenarios for combinatorial optimization problems. They are also commonly used in benchmarking GNN models, as in GraphWorld (Palowitch et al., 2022).

Interestingly, Bourgain also showed that random graphs are difficult to embed in Euclidean space while preserving distances (Bourgain, 1985, Section 3). For networks of size $n$, random graphs require an embedding dimension of $O(\log n)$ for a $O(\log n/\log\log n)$-factor approximation, close to the worst-case guarantee of $O(\log n)$. Studying the performance of local-global algorithms on random graphs for constant-factor approximations further motivates our work.

*Our main contribution is theoretical:* we show that local-global algorithms provide $(1-\varepsilon)$-factor lower bounds and $(1+\varepsilon)$-factor upper bounds for the shortest distances for most pairs of nodes with high probability. The proof leverages branching process approximations developed in the random graph literature (van der Hofstad, 2017; 2024).

In the worst-case setting, (Sarma et al., 2010), (Matoušek, 1996), and (Awasthi et al., 2022) showed that local-global algorithms can achieve a $(2c-1)$-factor upper bound and a $\frac{1}{2c-1}$-factor lower bound with an embedding dimension of $\Omega(n^{1/c}\log n)$ for $c > 1$. Our results on ER graphs requires an embedding dimension of $\Omega\left(n^{1/c}\log n\frac{1/c}{2\log 2}\right)$ for a tighter $\left(2-\frac{1}{c}\right)$-factor upper bound and $\Omega\left(n^{1/c}\log n\frac{1-1/c}{2\log 2}\right)$ for a tighter $\frac{1}{c}$-factor lower bound, achieving improved embedding dimension requirements for most node pairs in random graphs.

**Methodological and empirical contributions.** Building on GNN+, we enhance the local-global shortest distance algorithm inspired by Bourgain's theorem by incorporating a GNN to compute local embeddings. In the local step, the GNN is trained to compute shortest path distances from a random subset of nodes $S$ to all other nodes in the graph. The local embedding of each node is calculated as $d(u, S_i) = \min_{s \in S_i} d(u, s)$. In the global step, the distance between nodes $u$ and $v$ is lower bounded by $\max_i |d(u, S_i) - d(v, S_i)|$.

The use of GNNs in the local step is motivated by their demonstrated alignment with dynamic programming (DP). DP underlies many reasoning tasks, including shortest paths which can be solved using the Bellman-Ford algorithm. Recent works have shown that GNNs align well with DP, meaning their computation structures naturally reflect the algorithmic processes of tasks like shortest path computation, which improves learning efficiency and generalization (Xu et al., 2019b, Theorem 3.6). In (Dudzik and Veličković, 2022), this alignment has been theoretically quantified, suggesting that GNN architectures are particularly well-suited for reasoning tasks where DP plays a central role.

Our empirical results on ER graphs and benchmark datasets demonstrate that the GNN-augmented algorithm improves over the traditional BFS-based approach. Notably, we show that GNNs trained on small ER graphs can transfer effectively to downstream shortest path computation on real-world social networks. This underscores the importance of analyzing graph algorithms in the context of random graphs to inform their practical applications.

**Notation.** We consider undirected, unweighted and connected graphs $G = (V, E)$ where $V$, $|V| = n$, is the set of nodes and $E \subseteq V \times V$, $|E| = m$, is the set of edges. We define the one-hop neighborhood of node $u$ as $N(u) = \{v \in V \mid (u, v) \in E\}$. We often use the Bachmann–Landau asymptotic notation $o(1), O(1), \omega(1), \Omega(1), \Theta(1)$ etc. For a discrete set $X$, $|X|$ denotes its cardinality. Given a sequence of probability measures $(\mathbb{P}_n)_{n\geq 1}$, a sequence of events $(\mathcal{E}_n)_{n\geq 1}$ is said to hold *with high probability (w.h.p.)* if $\lim_{n\to\infty} \mathbb{P}_n(\mathcal{E}_n) = 1$. For a sequence of random variables $(X_n)_{n\geq 1}$,

$X_n \xrightarrow{\mathbb{P}} c$ means that $X_n$ converges to $c$ in probability. We write statements such as $X_n = f(n)^{o(1)}$ w.h.p. to abbreviate that $\log X_n / \log f(n) \xrightarrow{\mathbb{P}} 0$. Also, we write $X_n = O(1)$ w.h.p. to mean that $\mathbb{P}(X_n \geq K) \to 0$ for a sufficiently large $K$.

## 2 SHORTEST PATH PROBLEMS AND LOCAL-GLOBAL ALGORITHMS

Given graph $G = (V, E)$ and a pair of nodes $u, v \in V$, the shortest path problem consists of finding the path with the smallest number of edges between $u$ and $v$, and the number of edges in this path, or the *shortest path distance* between $u$ and $v$, denoted $d(u, v)$. This is one of the most fundamental combinatorial optimization problems on graphs.

The classical algorithm for finding graph shortest paths is Dijkstra's algorithm. Starting from a source node $u$, Dijkstra's algorithm returns the exact shortest paths between $u$ and every other node $v \in V$ along with the corresponding distances $d(u, v)$ via breadth-first search (BFS). It proceeds as follows:

(0) Initialize $d(u, v) = \infty$ for all $v \in V$. Set $s = u$ and $\Delta = 1$.

(1) From $s$, visit $s$'s neighbors $v \in N(s)$ and assign them distance $d(u, v) = \min(d(u, v), \Delta)$.

(2) Mark $s$ as visited and update $\Delta = \Delta + 1$.

(3) Select the unvisited node with smallest distance to $u$, say $t$, and set $s = t$.

(4) Repeat (1)–(3) until convergence.

Using naive data structures to store nodes' visited statuses and current distances, the complexity of Dijsktra's algorithm is $O(n^2)$. This can be improved to $O(m + n \log n)$ by using more efficient data structures like heaps Schrijver (2012), but is still prohibitive for large graphs.

### 2.1 LOWER AND UPPER BOUNDS ON SHORTEST PATH DISTANCE

While computing exact shortest path distances is expensive, we can afford to compute local paths. At a high level, local-global algorithms leverage this idea as follows. First, they sample a number of seed nodes that are stored in a set $S$. Then, for each node in $V$, they compute the shortest path distance to the nodes in $S$. This is the so-called local step, as in practice the shortest paths between $v \in V$ and $s \in S$ can be computed via BFS from $S$.

$$\text{Local step: Sample seed nodes } s \in S. \text{ Compute exact } d(s, v) \text{ for all } s \in S, v \in V. \quad (1)$$

Using the triangle inequality, the distances between the nodes in $S$ and $V$ can be used to approximate $d(u, v)$ for any $u, v \in V$. in two ways.

**Lower bound (LB).** Let $u, v \in V$, and $s \in S$. By the triangle inequality, we have $d(u, s) \leq d(u, v) + d(v, s)$, hence $d(u, v)$ can be lower bounded as

$$|d(u, s) - d(v, s)| \leq d(u, v)$$

since $d(u, s)$ and $d(v, s)$ are known from (1). For arbitrary $s$, this approximation is however very coarse. Therefore, in practice we search over all $s \in S$ and find the one that maximizes the left-hand-side. More formally, we can formulate this as follows. Given the exact distances $d(u, s_i)$ for all $u \in V$ and $s_i \in S$ for $i = 1, 2, .., |S|$, construct an embedding vector

$$\mathbf{x}_u = [d(u, s_1) \ldots d(u, s_{|S|})] \quad (2)$$

for each $u \in V$. Then, the best lower bound on $d(u, v)$ is given by $\|\mathbf{x}_u - \mathbf{x}_v\|_\infty$. This is the so-called global step, as the infinity norm requires taking the maximum over all vector entries.

$$\text{Global step for LB: Compute } \hat{d}(u, v) = \|\mathbf{x}_u - \mathbf{x}_v\|_\infty \text{ for all } u, v \in V. \quad (3)$$

**Upper bound (UB).** To find an upper bound $d(u, v)$, we can once again use the triangle inequality as

$$d(u, v) \leq d(u, s) + d(s, v).$$

Similarly to what we did for the lower bound, we want to pick the seed $s$ for which this upper bound is the tightest. Using the same embeddings $\mathbf{x}_u$ from (2), the global step is then

$$\text{Global step for UB: Compute } \tilde{d}(u, v) = \min_i [\mathbf{x}_u + \mathbf{x}_v]_i \text{ for all } u, v \in V. \quad (4)$$

## 2.2 Lower and Upper Bound Distortions, and an Algorithm that Achieves Them

The pseudoalgorithms defined by the local and global steps in (1),(3), (4) are only useful if we can derive guarantees on their approximation ability. For the LB, these can be obtained from Bourgain's classical embedding theorem, which characterizes the distortion incurred by optimal embeddings of metric spaces onto $\mathbb{R}^{|S|}$ equipped with the $\ell_\infty$ norm. For the UB, similar guarantees were derived in (Sarma et al., 2010).

**Theorem 2.1** (LB distortion, adapted from Matoušek (1996),Awasthi et al. (2022)). *Let $G$ be a graph with $n \geq 3$ nodes. Let $c > 1$. If $D = \Omega(n^{1/c} \log n)$, then there exist node embeddings $\mathbf{x}_u^* \in \mathbb{R}^D$, $u \in V$, for which $\hat{d}(u,v) = \|\mathbf{x}_u^* - \mathbf{x}_v^*\|_\infty$ satisfies*

$$\frac{d(u,v)}{2c-1} \leq \hat{d}(u,v) \leq d(u,v). \tag{5}$$

**Theorem 2.2** (UB distortion, Sarma et al. (2010)). *Let $G$ be a graph with $n \geq 3$ nodes. Let $c > 1$. If $D = \Omega(n^{1/c} \log n)$, then there exist node embeddings $\mathbf{x}_u^* \in \mathbb{R}^D$, $u \in V$, for which $\tilde{d}(u,v) = \min_i [\mathbf{x}_u^* + \mathbf{x}_v^*]_i$ satisfies*

$$d(u,v) \leq \tilde{d}(u,v) \leq (2c-1)d(u,v). \tag{6}$$

In order for (5) and (6) to hold, we need the embeddings $\mathbf{x}_u^*$ to be optimal. Yet, there is no guarantee that this is the case for the embeddings $\mathbf{x}_u$ in (2).

One way to ensure good embeddings is to control how we sample the seeds. Sarma et al. (2010) proposed a method for doing so that we describe in Algorithm 1. This method consists of first sampling $r + 1$ seed sets $S_0, S_1, \ldots, S_r$ of various sizes. Instead of recording distances of $u$ to every node in every set $S_i$, the embeddings only keep track of the minimum distance to the set, i.e., $[\mathbf{x}_u]_i = \min_{s \in S_i} d(u,s)$.

---

**Algorithm 1:** Local-Global Algorithm (adapted from Sarma et al. (2010))

---

**Input:** Graph $G = (V, E)$, $|V| = n$. Number of seed sets $r + 1$. Seed sets sizes $|S_i|$.
**Output:** Shortest path approximations $\hat{d}(u,v)$, $\tilde{d}(u,v)$ for all $u, v \in V$.
**for** $i = 0, 1, \ldots, r$ ;                                           /* Local step */
**do**
    $S_i \leftarrow \{s_1, \ldots, s_{|S_i|} \sim \text{Uniform}(V)\}$ ;
    **for** $u = 1, \ldots, n$ **do**
        $[\mathbf{x}_u]_i = \min_{s \in S_i} \text{Dijkstra}(s, u)$
        $[\boldsymbol{\sigma}_u]_i = \text{argmin}_{s \in S_i} \text{Dijkstra}(s, u)$
    **end**
**end**
**for** $u = 1, \ldots, n$ ;                                           /* Global step */
**do**
    **for** $v = 1, \ldots, n$ **do**
        $\hat{d}(u,v) = \|\mathbf{x}_u - \mathbf{x}_v\|_\infty$ ;                                           /* Lower bound */
        $\tilde{d}(u,v) = \min_i [(\mathbf{x}_u + \mathbf{x}_v) \odot \mathbf{1}(\boldsymbol{\sigma}_u = \boldsymbol{\sigma}_v)]_i$ ;                                           /* Upper bound */
    **end**
**end**

---

For the LB, smaller seed set sizes are beneficial as, for $k_1 + k_2 < 1$ with $k_1 < k_2$, we must find at least one seed set with a seed in the ball of radius $k_1 d(u,v)$ centered at $u$, and no seeds in the ball of radius $k_2 d(u,v)$ centered at $v$. Hence, having a range of seed set sizes helps.

For the UB, this strategy ensures a seed falls at the intersection of the $\lceil \frac{d(u,v)}{2} \rceil$-hop neighborhoods of nodes $u$ and $v$ w.h.p. In this case, an auxiliary vector $\boldsymbol{\sigma}_u$ is also defined to store the index of the closest node to $u$ in the set $S_i$, i.e., $[\boldsymbol{\sigma}_u]_i = \text{argmin}_{s \in S_i} d(u,s)$. This method is described in detail in Algorithm 1[1].

---

[1]$\mathbf{1}(\cdot)$ denotes the elementwise Boolean function.

It can be shown that if $r = \lfloor \log n \rfloor$, the $|S_i|$ are exponential in $i$, and the local step of Algorithm 1 is run for $R = \Theta(n^{1/c})$ rounds—yielding a total embedding size of $\Theta(n^{1/c} \log n)$—, the resulting shortest path distance approximations satisfy Theorems 2.1 and 2.2 with high probability for *any graph*. In the following, we show that the distortion and the embedding dimension can both be improved for random graphs.

# 3 LOWER AND UPPER BOUND DISTORTIONS ON SPARSE ERDŐS-RÉNYI GRAPHS

In this section, we will state and prove our main results concerning performance of Algorithm 1 on a sparse Erdős-Rényi graph. An ER random graph model $\mathrm{ER}_n(\lambda/n)$ generates a random graph on $n$ nodes, and each edge $\{i, j\}$ is included in the graph with probability $\frac{\lambda}{n}$, independently. Thus, $\mathrm{ER}_n(\lambda/n)$ is a distribution over the space of all graphs on $n$ nodes. We write $G \sim \mathrm{ER}_n(\lambda/n)$ to abbreviate that $G$ is distributed as $\mathrm{ER}_n(\lambda/n)$. Let $C_{(i)}$ be the $i$-th largest connected component in an ER graph. A well-known result in the theory of random graphs (cf. (van der Hofstad, 2017, Theorems 4.4, 4.8, and Corollary 4.13)) is the existence of a unique giant component in an ER graph, which states the following:

**Theorem 3.1** (van der Hofstad (2017)). *Let $G \sim \mathrm{ER}_n(\lambda/n)$ and $C_{(i)}$ be as defined above. If $\lambda < 1$, then $\frac{C_{(1)}}{n} = O\left(\frac{\log n}{n}\right)$ w.h.p. On the other hand, if $\lambda > 1$, then $\frac{C_{(1)}}{n} \xrightarrow{\mathbb{P}} \zeta$ for some $\zeta > 0$ and $\frac{C_{(2)}}{n} = O\left(\frac{\log n}{n}\right)$ w.h.p.*

Throughout, we will consider a fixed $\lambda > 1$, since otherwise $\mathbb{P}(u_1, u_2$ lie in the same component$) \to 0$ as $n \to \infty$, for any $u_1, u_2$.

## 3.1 LOWER BOUND DISTORTION

On ER graphs, we obtain the following distortion result for the lower bound $\hat{d}(u, v)$ in Algorithm 1.

**Theorem 3.2.** *Let $G \sim \mathrm{ER}_n(\lambda/n)$ and let $u_1, u_2$ be two nodes chosen independently and uniformly at random with replacement (the choice of $u_1, u_2$ is also independent of $G$). Fix $\varepsilon \in (0, 1)$. Let $\hat{d}(u_1, u_2)$ be the output of Algorithm 1 for the lower bound on the shortest distance $d(u_1, u_2)$ after $R = \omega(n^{1-\varepsilon})$ runs of the local step, with $|S_i| = 2^i$ for $i = 0, 1, .., r$ and $r = \lfloor \log n \frac{\varepsilon}{2 \log 2} \rfloor$, yielding node embedding dimension $D = \Omega\left(n^{1-\varepsilon} \log n \frac{\varepsilon}{2 \log 2}\right)$. Then, with high probability, $\hat{d}(u_1, u_2) \geq (1 - \varepsilon)d(u_1, u_2)$, i.e., $\hat{d}(u_1, u_2)$ provides a $(1 - \varepsilon)$-approximation of $d(u_1, u_2)$.*

**Idea of the proof.** Let $N_k(u)$ denote the set of nodes with graph distance at most $k$ from $u$ and $\partial N_k(u)$ denote the set of nodes with graph distance exactly $k$ from $u$. The first part of the proof relies on local neighborhood expansions of ER random graphs. In particular, the boundaries of the $k$-th neighborhoods of $u_1, u_2$ grow exponentially as $\lambda^k$. This is a consequence of the following intermediate result.

**Lemma 3.3.** *Let $G$, $u_1$, and $u_2$ be as in Theorem 3.2. Let $L = \kappa_0 \log_\lambda n$ with $\kappa_0 \in (0, \frac{1}{2})$ and $\varepsilon > 0$ be sufficiently small. Let $\mathcal{A}_{b_1, b_2}$ be the event that $|\partial N_L(u_i)| = b_i$ for $i = 1, 2$ where $b_i \in (n^{-\varepsilon} \lambda^L, n^\varepsilon \lambda^L)$. Fix $\kappa \in (0, 1 - \kappa_0)$, and let $\mathcal{E}_n$ be the good event that $|\partial N_{k_i}(u_i)| \in (n^{-\varepsilon} \lambda^{k_i}, n^\varepsilon \lambda^{k_i})$ for all $k_i \leq (\kappa + \kappa_0) \log_\lambda n$ and $i = 1, 2$. Then, there exists $\delta > 0$ such that $\mathbb{P}(\mathcal{E}_n \mid \mathcal{A}_{b_1, b_2}) \geq 1 - n^{-\delta}$ for all sufficiently large $n$.*

*Proof.* See Appendix B.

To find the $(1 - \varepsilon)$-factor lower bound for $\hat{d}(u_1, u_2)$ when $u_1, u_2$ are in the same connected component, we consider two disjoint balls centered at $u_1$ and $u_2$ with radii differing by a factor of $1 - \varepsilon$. If there exists a seed set that contains at least one point in the ball of smaller radius and is disjoint from the ball of larger radius, then $\hat{d}(u_1, u_2)$ returned by Algorithm 1 is lower bounded by the larger radius minus the smaller radius. The complete proof of Theorem 3.2 can be found in Appendix A.

## 3.2 UPPER BOUND DISTORTION

On ER graphs, we obtain the following distortion result for the upper bound $\tilde{d}(u, v)$ in Algorithm 1.

**Theorem 3.4.** *Let $G$, $u_1$, and $u_2$ be as in Theorem 3.2. Fix $\varepsilon \in (0, 1)$. Let $\tilde{d}(u_1, u_2)$ be the output of Algorithm 1 for the upper bound on the shortest distance $d(u_1, u_2)$ after $R = \omega(n^{1-\varepsilon})$ runs of the local step, with $|S_i| = 2^i$ for $i = 0, 1, .., r$ and $r = \lfloor \log n \frac{1-\varepsilon}{2 \log 2} \rfloor$, yielding node embedding dimension $D = \Omega\left(n^{1-\varepsilon} \log n \frac{1-\varepsilon}{2 \log 2}\right)$. Then, with high probability, $\tilde{d}(u_1, u_2) \leq (1 + \varepsilon)d(u_1, u_2)$, i.e., $\tilde{d}(u_1, u_2)$ provides a $(1 + \varepsilon)$-approximation of $d(u_1, u_2)$.*

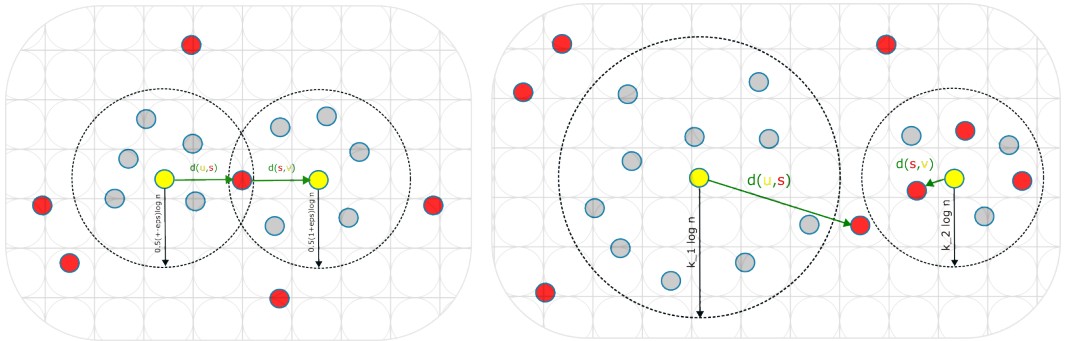

Figure 1: Schematics depicting the computation of the upper bound (left) and lower bound (right). Yellow nodes are the source and target, red nodes are seeds in a seed set $S$, and gray nodes are arbitrary nodes. **Left:** To achieve the upper bound in Theorem 3.4, only one seed can be at the union of the balls around $u$ and $v$, and it must lie at the intersection. The need for lying at the intersection is clear; that gives us our shortest path approximation. Having only one seed come from the union ensures the algorithm will output at most $\tilde{d}(u, v) = d(u, S) + d(v, S) \leq (1 + \varepsilon)d(u, v)$ w.h.p. **Right:** The balls around $u$ and $v$ are disjoint and we consider $k_1 \geq k_2 + 1 - \varepsilon$. To achieve the lower bound in Theorem 3.2, at least one seed must lie on the ball around $v$, and no seed can lie on the ball around $u$. This guarantees at least $\hat{d}(u, v) = d(u, S) - d(v, S) \geq (1 - \varepsilon)d(u, v)$ w.h.p.

**Idea of the proof.** Let $N_k(u)$ denote the set of nodes with graph distance at most $k$ from $u$. The fact that the boundaries of the $k$-th neighborhoods of $u_1, u_2$ grow exponentially as $\lambda^k$ (Lemma 3.3) allows us to show that $|N_k(u_1) \cup N_k(u_2)|$ grow as $\lambda^k$ and $|N_k(u_1) \cap N_k(u_2)|$ grow as $\frac{\lambda^{2k}}{n}$. This is formalized in the following proposition.

**Proposition 3.5.** *Let $G$, $u_1$, and $u_2$ be as in Theorem 3.2. Let $\varepsilon > 0$ be sufficiently small. For any $\kappa_0 \log_\lambda n \leq k \leq (\kappa + \kappa_0) \log_\lambda n$ with $\kappa_0 \in (0, \frac{1}{2})$ and $\kappa \in (0, 1 - \kappa_0)$, the following holds with high probability, conditionally on $u_1, u_2$ being in the same connected component:*

$$|N_k(u_1) \cap N_k(u_2)| \in \left(n^{-\varepsilon} \frac{\lambda^{2k}}{n}, n^\varepsilon \left(\frac{\lambda^{2k}}{n} + 1\right)\right) \quad \text{and} \quad |N_k(u_1) \cup N_k(u_2)| \in \left(n^{-\varepsilon} \lambda^k, n^\varepsilon \lambda^k\right).$$

*Proof.* See Appendix D. $\square$

**Lemma 3.6.** *Let $\varepsilon > 0$ be sufficiently small and let $L = \kappa_0 \log_\lambda n$ with $\kappa_0 \in (0, \frac{1}{2})$. Let $A_n$ denote the event that $n^{-\varepsilon} \lambda^L \leq |\partial N_L(u_i)| \leq n^\varepsilon \lambda^L$ for $i = 1, 2$ and $B_n$ denote the event that $u_1$ and $u_2$ are in the same connected component. Then $\mathbb{P}(A_n \setminus B_n) \to 0$ and $\mathbb{P}(B_n \setminus A_n) \to 0$ as $n \to \infty$.*

*Proof.* See Appendix E.

Given these growth rates, the main idea is to show that, with high probability, there exists a seed set $S_i$ such that it has exactly one seed in $N_k(u_1) \cup N_k(u_2)$ that also lies in $N_k(u_1) \cap N_k(u_2)$, where $k = \frac{1}{2}(1 + \varepsilon)d(u_1, u_2)$. Thus, with high probability, the output $\tilde{d}(u, v)$ of Algorithm 1 is at most sum of distances from $u_1$ to $S_i$ and $u_2$ to $S_i$. Due to the choice of $k$, the output is therefore at most $(1 + \varepsilon)d(u_1, u_2)$. The complete proof of Theorem 3.4 can be found in Appendix C.

# 4 A GNN-BASED ALGORITHM AND EXPERIMENTAL RESULTS

We propose to modify Algorithm 1 by implementing the local step with a GNN. GNNs are deep convolutional architectures tailored to graph data Scarselli et al. (2008); Kipf and Welling (2017); Defferrard et al. (2016); Ruiz et al. (2021). Specifically, we focus on node data that we represent as matrices $\mathbf{X} \in \mathbb{R}^{n \times d}$. Each row of $\mathbf{X}$ corresponds to a node $u \in V$ and each column to a different signal or feature. A GNN layer operates on this type of data via two operations: a graph convolution and a pointwise nonlinearity. Explicitly, let $\mathbf{X}_{\ell-1} \in \mathbb{R}^{n \times d_{\ell-1}}$ be the input to layer $\ell$ (or equivalently the output of layer $\ell - 1$). The $\ell$th layer is given by

$$\mathbf{X}_\ell = \sigma \left( \sum_{k=0}^{K-1} \mathbf{A}^k \mathbf{X}_{\ell-1} \mathbf{W}_{\ell,k} \right) \tag{7}$$

where $\mathbf{A} \in \mathbb{R}^{n \times n}$ is the graph adjacency, $\mathbf{W}_{\ell,k} \in \mathbb{R}^{d_{\ell-1} \times d_\ell}$ are learnable parameters and $\sigma$ is a pointwise nonlinear activation function such as the ReLU or sigmoid. A GNN stacks $L$ such layers, the first layer input $\mathbf{X}_0$ being the input data $\mathbf{X}$ and the last layer output $\mathbf{X}_L$ the output data $\mathbf{Y}$. To be concise, we represent the entire GNN as a map $\mathbf{Y} = \Phi(\mathbf{X}, \mathbf{A}; \mathcal{W})$ where $\mathcal{W} = \{\mathbf{W}_{\ell,k}\}_{\ell,k}$ groups the learnable parameters across all layers.

An important property GNNs inherit from graph convolutions is locality. More specifically, the operations involved in each GNN layer can be implemented locally at each node via one-hop information exchanges with their neighbors. To see this, consider a one-dimensional signal $\mathbf{x} \in \mathbb{R}^n$. The operation $\mathbf{z} = \mathbf{A}\mathbf{x}$ is local in the sense that

$$[\mathbf{z}]_u = [\mathbf{A}\mathbf{x}]_u = \sum_{v \in V} [\mathbf{A}]_{uv} [\mathbf{x}]_v = \sum_{v \in N(u)} [\mathbf{A}]_{uv} [\mathbf{x}]_v$$

where $N(u)$ is the neighborhood of node $u$. Similarly, $\mathbf{z}_k = \mathbf{A}^k \mathbf{x}$ can be implemented locally in $R$ rounds by unrolling $\mathbf{z}_k = \mathbf{A}\mathbf{z}_{k-1}$. The nonlinearity $\sigma$ is pointwise and hence also local.

Leveraging the locality property of GNNs, we replace the local step of Algorithm 1 by a GNN forward pass. Instead of calculating the embeddings $\mathbf{x}_u$ via BFS, we propose to learn them using a GNN.

*Remark* 4.1. The motivation for using GNNs in the local step is threefold. First, once the GNN is trained, the sketch computations become automated. Second, by using GNNs we can save computations as, if $L < \log n$, GNN inference is cheaper than BFS on ER graphs. Third, we can leverage the GNN transferability property Ruiz et al. (2020; 2023) to transfer the learned model to graphs of different sizes associated with the same graph model.

## 4.1 EXPERIMENT 1: LEARNING THE GNN

In order to train the GNN, we proceed as follows. We sample a training set of ER graphs with $n$ nodes and generate random input signals $\mathbf{X} \in \mathbb{R}^{n \times r}$ satisfying $\mathbf{1}_n^T \mathbf{X} \mathbf{1}_r = r$ and $\mathbf{1}_n^T \mathbf{X} = \mathbf{1}_r^T$. I.e., each column corresponds to a seed and one-hot encodes which node is a seed for a given graph. The outputs have the same dimensions as the inputs, $\mathbf{Y} \in \mathbb{R}^{n \times r}$, and correspond to the shortest path distances between nodes $u \in V$ and seeds $s \in S$, i.e., $[\mathbf{Y}]_{us} = d(u,s)$.

Before testing our algorithm, we assess the ability of the learned GNNs to compute end-to-end shortest paths. For this experiment, we consider $n = 50$, two values of $\lambda$, and set the GNN depth to $\lceil \log_\lambda n \rceil$. The results of this experiment are shown in Figure 2, where we plot the actual shortest path distance versus the shortest path distances predicted by four different GNN architectures (GCN Kipf and Welling (2017), GraphSAGE Hamilton et al. (2017), GAT Veličković et al. (2018), and GIN Xu et al. (2019a)). We observe that the GNN predictions saturate in both plots, signaling the inability of the GNN to predict longer distances even when their depth is higher than the expected path length of $\log_\lambda n$. As expected, GNNs are not suitable for computing end-to-end shortest path distances, especially on sparser graphs ($\lambda = 4$), which tend to exhibit longer shortest paths.

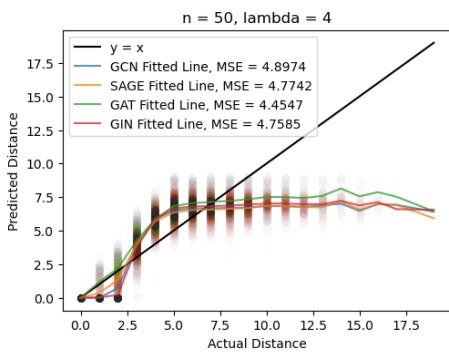
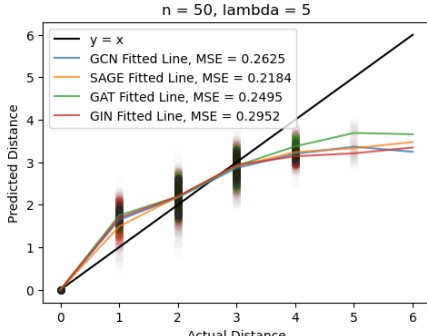

Figure 2: Raw outputs of $\lfloor\sqrt{n}\rfloor$-64-32-16-$\lfloor\sqrt{n}\rfloor$ GNNs that are trained on ER graphs generated by $ER_n(\lambda/n)$ with $\lambda \in \{4,5\}$ to predict shortest path distances end-to-end. Evaluation data are ER graphs from the same model.

## 4.2 EXPERIMENT 2: COMPARISON WITH ALGORITHM 1

Next, we evaluate the difference between the lower bound achieved using the GNN-based algorithm, and the lower bound from Algorithm 1. *Only lower bounds are compared to ensure a fair comparison, as the upper bound computation requires storing additional information—-the index of the closest seed in a seed set to each node.*

For this experiment, we consider a range of values of $n$. We limit the GNN depth to $L \ll log_\lambda n$ and tune $L$ and other parameters via cross-validation; see Appendix F for details. We also allow for $R$ rounds of the local step, i.e., we sample $r + 1$ seed sets as defined in Algorithm 1 in $R$ rounds, and save all $R(r + 1)$ distances to use in the global step.

The results of this experiment are shown in Figure 3 for $\lambda = 4$ and $\lambda = 5$. The GNN lower bound is worse than the vanilla lower bound on the $\lambda = 4$ graph, though it leads to a substantial improvement on the $\lambda = 5$ graph for all values of $R$. While both values of $\lambda$ correspond to the supercritical regime ($\lambda > 1$), there are a few factors explaining the difference in these two cases. As we could see from Figure 2, the GNN learns much worse local embeddings in the $\lambda = 4$ case, even for a small 50-node graph. Furthermore, for large values of $n$ the graph is almost surely connected when $\lambda = 5$, but not when $\lambda = 4$. This is an important distinction which can also be observed from the worsening of the GNN-based algorithm performance at $n \approx 100$ for $\lambda = 4$ (note that for $n < 100$, $4 > \log_4 n$). Finally, the GNN-based algorithm is faster than Algorithm 1, especially on large graphs, which is expected as exact local sketch computations via Dijkstra's algorithm scale poorly with the graph size.

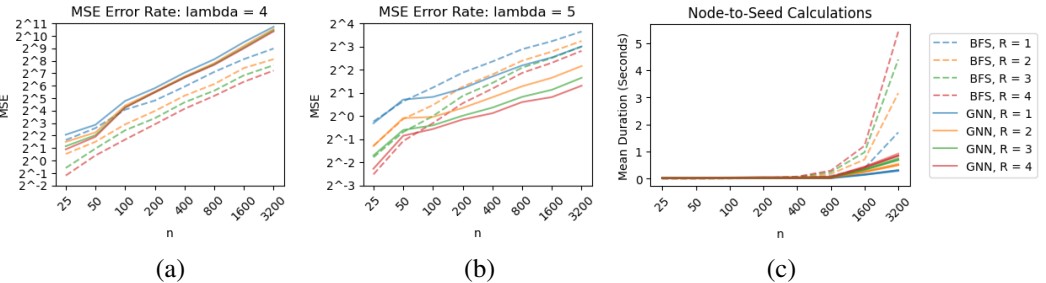

(a)                                    (b)                                    (c)

Figure 3: (a)-(b) Performance of BFS-based embeddings vs. GNN-based embeddings with GNNs trained on ER graphs generated by $ER_n(\lambda/n)$ for $\lambda \in \{4,5\}$. (c) Time required to generate all node-to-seed distances in ER graphs with $n$ nodes by NetworkX's highly optimized BFS as compared with our widest and deepest GNNs. All GCN-, GraphSage-, GAT-, and GIN-based algorithms are represented by the same color and line style for the same $R$, and the deviations between them are insignificant.

## 4.3 EXPERIMENT 3: TRANSFERABILITY

In our last experiment, we examine whether we can transfer GNNs learned on small graphs to compute local embeddings on larger networks, and use these embeddings for downstream approximation of shortest paths on these larger networks. This is motivated by theoretical and empirical work Ruiz et al. (2020; 2023) showing that GNNs are transferable in the sense that their outputs converge on convergent graph sequences, which in turn implies that they can be trained on smaller graphs and transferred to larger (but similar) graphs.

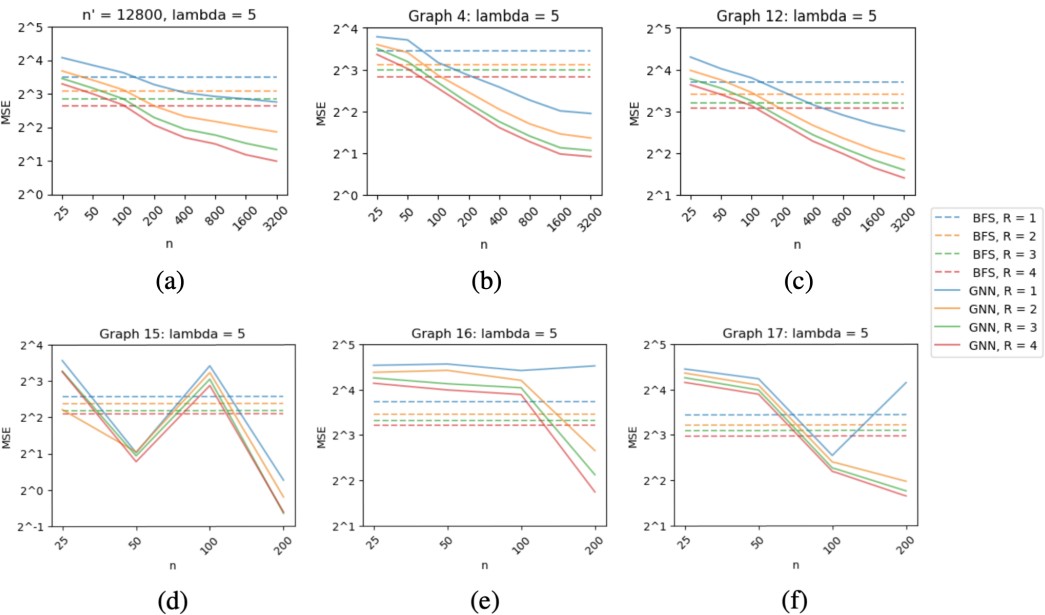

(a)     (b)     (c)

(d)     (e)     (f)

Figure 4: Error rates on (a) test ER graphs generated by $ER_{n'}(\lambda/n')$, (b) GEMSEC company network with 14,113 nodes, (c) Arxiv HEP-TH collaboration network with 28,281 nodes, (d) GEMSEC artist network with 41,618 nodes, (e) ER-AVGDEG10-100K-L2 labeled network with 99,997 nodes, and (f) Brightkite social network with 56,739 nodes by BFS-based embeddings vs. GNN-based embeddings using GNNs trained on ER graphs from $ER_n(\lambda/n)$ with $\lambda = 5$. See the Appendix for further details and more transferability results on real-world networks.

Here, we focus on the $\lambda = 5$ case and train a sequence of eight GNNs on graphs ranging from $n = 25$ to $n = 3200$ nodes. Then, we use these GNNs to compute shortest path distances using our GNN-based algorithm on an ER graph with same $\lambda$ and graph size of $n' = 12800$ nodes (additional experiment details are provided in Appendix F). Figure 4 (a) shows the MSE achieved in each instance with respect to the true shortest path distances as a function of the training graph size, with the flat dashed lines representing the MSE achieved by Algorithm 1 on the $n'$-node graph. We observe a steady decrease of the MSE as $n$ increases, and that the GNN-based algorithm matches the performance of Algorithm 1 when the GNN is trained on graphs of $n = 100$ nodes—which is 128 times smaller than the target $n'$-node graph.

We also examine the transferability of the same set of GNNs to seventeen real-world networks with sizes ranging from 3,892 to 99,997 nodes and average degrees between 4.19 and 26.77. In certain scenarios, random graphs can be used to model social networks Newman and Watts (1999). Therefore, we hypothesize that GNNs trained on ER graphs should produce good quality embeddings for these networks. The results on five of the seventeen networks are shown in Figure 4 (b-f), where we once again observe MSE improvement with the training graph size and that the GNN-based algorithm outperforms Algorithm 1 even when the embeddings are learned on much smaller graphs. The results on the remaining twelve real-world networks are provided in the Appendix.

## 5 CONCLUSION

We introduce an average-case analysis of algorithms combining local and global computations for solving shortest distance problems on ER graphs, complementing Bourgain's worst-case result. In particular, our theoretical analysis demonstrates that on ER graphs these algorithms can achieve a $(1 - \varepsilon)$-factor lower bound and a $(1 + \epsilon)$-factor upper bound of shortest distances with high probability. Additionally, we propose a modification to Bourgain's algorithm, which incorporates GNNs in the local computation phase to further enhance practical performance. Empirical results on both ER graphs and benchmark datasets demonstrate the superior performance of the GNN-augmented approach.

**Limitations and future work.** Our analysis focuses on ER random graphs, which provided a simplified framework to develop theoretical tools and insights for local-global algorithms. These methods are broadly applicable to graphs with local expansion properties, such as inhomogeneous random graphs, and extending our analysis to such models is a key direction for future work. However, for other important graph classes in shortest path problems, such as planar graphs, our techniques are unlikely to apply. Addressing these cases will require the development of new methods, which is an interesting direction for future work.

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

## A   PROOF OF THEOREM 3.2

We show that, for any $\varepsilon \in (0,1)$, $\hat{d}(u_1, u_2) \geq (1-\varepsilon)\log_\lambda n$ w.h.p. Let $S_{ij}$ be the $i$-the seed set having size $2^i$ in the $j$-th round. Let $k_1 = \varepsilon_1 \log_\lambda n$ and $k_2 = (1-\varepsilon)\log_\lambda n + k_1$, where $\varepsilon_1 \in \left(0, \frac{\varepsilon}{2}\right)$ (to be chosen later) (we let $\varepsilon_1 < \frac{\varepsilon}{2}$ so that an argument identical to (16) would yield that $N_{k_1}(u_1) \cap N_{k_2}(u_2) = \varnothing$ w.h.p., conditionally on $u_1, u_2$ being in the same connected component). Define $Z_{ij}$ to be the event that $S_{ij} \cap N_{k_1}(u_1) \neq \varnothing$ but $S_{ij} \cap N_{k_1}(u_2) = \varnothing$. Note that, if $Z_{ij}$ happens for some $i \leq r, j \leq R$, then $d(u_1, S_{ij}) \leq k_1$ and $d(u_2, S_{ij}) \geq k_2$, and consequently, $\hat{d}(u_1, u_2) \geq (1-\delta)\log_\lambda n$. Thus, denoting $Z = \cup_{i \leq r, j \leq R} Z_{ij}$, it suffices to prove that $\mathbb{P}(Z \mid G, u_1, u_2)\mathbb{1}\{u_1 \leftrightarrow u_2\} \xrightarrow{\mathbb{P}} 1$, where $u_1 \leftrightarrow u_2$ stands for $u_1, u_2$ being in the same connected component. Let $\mathcal{C}_{(i)}$ be the set of nodes in the $i$-th largest connected component of $G$ and $C_{(i)}$ be the number of nodes in $\mathcal{C}_{(i)}$. Note that, $\mathbb{P}(u_1 \leftrightarrow u_2,$ but $u_1, u_2 \notin \mathcal{C}_{(1)} \mid G) = \frac{1}{n^2}\sum_{i \geq 2}|C_{(i)}|^2 \leq \frac{C_{(2)}}{n} \xrightarrow{\mathbb{P}} 0$. Therefore, $\mathbb{P}(\{u_1 \leftrightarrow u_2\} \triangle \{u_1, u_2 \in \mathcal{C}_{(1)}\}) \xrightarrow{\mathbb{P}} 0$, where $\triangle$ denotes the symmetric difference between sets. Thus, it suffices to show that $\mathbb{P}(Z \mid G, u_1, u_2)\mathbb{1}\{u_1, u_2 \in \mathcal{C}_{(1)}\} \xrightarrow{\mathbb{P}} 1$ (or equivalently $\mathbb{P}(Z^c \mid G, u_1, u_2)\mathbb{1}\{u_1, u_2 \in \mathcal{C}_{(1)}\} \xrightarrow{\mathbb{P}} 0$).

The fact that $\mathbb{P}(A^c \cap B) = \mathbb{P}(B) - \mathbb{P}(A \cap B)$ implies, for each $i, j$,

$$\mathbb{P}(S_{ij} \cap N_{k_1}(u_1) \neq \varnothing, S_{ij} \cap N_{k_2}(u_2) = \varnothing \mid G, u_1, u_2)$$
$$= \left(1 - \frac{|N_{k_2}(u_2)|}{n}\right)^{2^i} - \left(1 - \frac{|N_{k_1}(u_1)| + |N_{k_2}(u_2)|}{n}\right)^{2^i}.$$

Therefore,

$$\mathbb{P}(Z^c \mid G, u_1, u_2) = \left(\prod_{i=0}^{r}\left(1 - \left(1 - \frac{|N_{k_2}(u_2)|}{n}\right)^{2^i} + \left(1 - \frac{|N_{k_1}(u_1)| + |N_{k_2}(u_2)|}{n}\right)^{2^i}\right)\right)^R$$

$$\leq \exp\left(-R\sum_{i=0}^{r}\left(\left(1 - \frac{|N_{k_2}(u_2)|}{n}\right)^{2^i} - \left(1 - \frac{|N_{k_1}(u_1)| + |N_{k_2}(u_2)|}{n}\right)^{2^i}\right)\right)$$

$$= \exp\left(-R\sum_{i=0}^{r}\frac{|N_{k_1}(u_1)|}{n}\sum_{j=0}^{2^i-1}\left(1 - \frac{|N_{k_2}(u_2)|}{n}\right)^{2^i-1-j}\left(1 - \frac{|N_{k_1}(u_1)| + |N_{k_2}(u_2)|}{n}\right)^j\right)$$

where the first "$\leq$" uses $1 - x \leq \exp(-x)$. By Lemma 3.3, $|N_{k_1}(u_1)| = n^{o(1)}\frac{\lambda n^{\varepsilon_1}-1}{\lambda-1} \leq n^{o(1)}\frac{\lambda n^{\varepsilon_1+1-\varepsilon}-1}{\lambda-1} = |N_{k_1}(u_1)| =$ w.h.p. Then with high probability,

$$\mathbb{P}(Z^c \mid G, u_1, u_2)\mathbb{1}\{u_1, u_2 \in \mathcal{C}_{(1)}\} \leq \exp\left(-R\frac{|N_{k_1}(u_1)|}{n}\sum_{i=0}^{r}2^i\left(1 - 2\frac{|N_{k_2}(u_2)|}{n}\right)^{2^i}\right)$$

$$\leq \exp\left(-Rn^{\varepsilon_1-1+o(1)}\sum_{i=0}^{r-1}2^i\left(1 - \frac{2\lambda}{\lambda-1}n^{\varepsilon_1-\varepsilon+o(1)}\right)^{\frac{2^r}{r}i}\right)$$

$$= \exp\left(-Rn^{\varepsilon_1-1+o(1)}\frac{2^r\left(1 - \frac{2\lambda}{\lambda-1}n^{\varepsilon_1-\varepsilon+o(1)}\right)^{2^r} - 1}{2\left(1 - \frac{2\lambda}{\lambda-1}n^{\varepsilon_1-\varepsilon+o(1)}\right)^{\frac{2^r}{r}} - 1}\right).$$

Since $1 \geq \left(1 - \frac{2\lambda}{\lambda-1} n^{\varepsilon_1 - \varepsilon + o(1)}\right)^{\frac{2^r}{r}} \geq \left(1 - \frac{2\lambda}{\lambda-1} n^{\varepsilon_1 - \varepsilon + o(1)}\right)^{2^r} \geq \left(1 - \right.$

$\left. \frac{2\lambda}{\lambda-1} n^{\varepsilon_1 - \varepsilon + o(1)}\right)^{2^{\log n \frac{\varepsilon}{2 \log 2}}} \to 1$ as $n \to \infty$, with high probability,

$$\mathbb{P}(Z^c \mid G, u_1, u_2) \mathbb{1}\{u_1, u_2 \in \mathcal{C}_{(1)}\} \leq \exp\left(-R n^{\varepsilon_1 - 1 + o(1)} c 2^r\right) \text{ (with } c \in (0,1))$$

$$< \exp\left(-R n^{\varepsilon_1 - 1 + o(1)} c 2^{\log n \frac{\varepsilon}{2 \log 2} - 1}\right) = \exp\left(-R \frac{c}{2} n^{\varepsilon_1 - 1 + \frac{\varepsilon}{2} + o(1)}\right)$$

Choosing $\varepsilon_1 = \frac{\varepsilon}{2} - o(1)$, $R = \omega(n^{1-\varepsilon})$ is sufficient for the final bound to tend to 0.

# B  PROOF OF LEMMA 3.3

The proof is adapted from (van der Hofstad, 2024, Section 2.6.4). Since we need an exponential bound on the probability and $L$ is growing with $n$, the proof does not follow from van der Hofstad (2024).

We start by proving that there exists a $\gamma \in (0,1)$ and $\delta' > 0$ such that for all sufficiently large $n$,

$$|N_k(u_i)| \leq n^\gamma, \quad \text{for all } k \leq (\kappa + \kappa_0) \log_\lambda n \text{ and } i = 1, 2, \tag{8}$$

with probability at least $1 - n^{-\delta'}$. Indeed, for any $r \geq 1$ and node $u$, $|\partial N_r(u)| = \sum_{i \in \partial N_{r-1}(u)} \sum_{j \notin N_{r-1}(u)} I_{ij}$, where $I_{ij}$ is the indicator random variable for the edge $\{i, j\}$ being present. Therefore, $\mathbb{E}[|\partial N_r(u)|] \leq \lambda E[|\partial N_{r-1}(u)|]$. Proceeding inductively, we have that $\mathbb{E}[|\partial N_r(u)|] \leq \lambda^r$ and consequently, $\mathbb{E}[|N_r(u)|] \leq \frac{\lambda^{r+1}-1}{\lambda-1} = O(\lambda^r)$. Since $\kappa_0 + \kappa < 1$, we can apply Markov's inequality to conclude that $|N_k(u_i)| \leq n^\gamma$ with probability $1 - n^{-\delta'}$ for some fixed $\delta' > 0$ and for any fixed $k \leq (\kappa + \kappa_0) \log_\lambda n$. Moreover, since $|N_k(u_i)| \leq |N_{k+1}(u_i)|$ for all $k$, we can conclude (8).

Next, fix $\varepsilon > 0$ sufficiently small and suppose that $\delta_n = n^{-\kappa_0/4}$. Define the event

$$\mathcal{E}_{k,(1)} := \left\{b_1[\lambda(1-\delta_n)(1 - n^{-(1-\gamma)})]^k \leq |\partial N_{L+k}(u_1)| \leq b_1[\lambda(1+\delta_n)]^k\right\}$$

We will upper bound $\mathbb{P}(\mathcal{E}_{k,(1)}^c \mid \cap_{l=0}^{k-1} \mathcal{E}_{l,(1)}, \mathcal{A}_{b_1,b_2})$. Again, using $|\partial N_{L+k}(u_1)| = \sum_{i \in \partial N_{L+k-1}(u_1)} \sum_{j \notin N_{L+k-1}(u_1)} I_{ij}$, we have that

$$E_n := \mathbb{E}[|\partial N_{L+k}(u_1)| \mid N_{L+k-1}(u_1), \mathcal{A}_{b_1,b_2}] = |\partial N_{L+k-1}(u_1)|(n - |N_{L+k-1}(u_1)|)\frac{\lambda}{n}. \tag{9}$$

Using (8), it follows that, with probability at least $1 - n^{-\delta'}$,

$$\lambda|\partial N_{L+k-1}(u_1)|(1 - n^{-(1-\gamma)}) \leq E_n \leq \lambda|\partial N_{L+k-1}(u_1)|. \tag{10}$$

Conditionally on $\cap_{l=0}^{k-1} \mathcal{E}_l$ and $\mathcal{A}_{b_1,b_2}$, with probability at least $1 - n^{-\delta'}$,

$$b_1\lambda^k(1-\delta_n)^{k-1}(1 - n^{-(1-\gamma)})^k \leq E_n \leq b_1\lambda^k(1+\delta_n)^{k-1}. \tag{11}$$

Using Standard concentration inequalities for sums of Bernoulli random variables (Janson et al., 2000, Theorem 2.8 and Corollary 2.3, (2.9)), we conclude that

$$\mathbb{P}(\mathcal{E}_{k,(1)}^c \mid \cap_{l=0}^{k-1} \mathcal{E}_{l,(1)}, \mathcal{A}_{b_1,b_2})$$

$$= \mathbb{P}(|\partial N_{L+k}(u_1) - E_n| > \delta_n E_n \mid \cap_{l=0}^{k-1} \mathcal{E}_l, \mathcal{A}_{b_1,b_2})$$

$$\leq 2e^{-\frac{\delta_n^2}{3} \times E_n} + n^{-\delta'} \leq n^{-\delta'/2}.$$

Therefore, $\mathbb{P}(\cap_{k \leq \kappa_0 \log_\lambda n} \mathcal{E}_k \mid \mathcal{A}_{b_1,b_2}) \geq 1 - n^{-\delta}$ for all sufficiently large $n$, for some $\delta > 0$. Finally, the proof follows by noting that $(1-\delta_n)^k = (1 - n^{-\kappa_0/4})^k \to 1$ and $(1 - n^{-(1-\gamma)})^k \to 1$ uniformly over $k \leq \kappa \log n$. An identical argument can be repeated for neighborhoods of $u_2$. In the latter case, we need to additionally condition on the $L + k$ neighborhood of $u_1$. With probability at least $1 - n^{-\delta'}$, this will result in exploration of at most $n^\gamma$ many nodes due to (8), and therefore, the asymptotics above also hold for neighborhoods of $u_2$. We skip redoing the proof for the neighborhoods of $u_2$ here.

## C  PROOF OF THEOREM 3.4

Let $k = \frac{1}{2}(1 + \varepsilon) \log_\lambda n$, where $\varepsilon \in (0, 1)$. For each fixed $i = 0, 1, \ldots, r$ and $j = 1, 2, \ldots, R$, let $S_{ij}$ be the seed set of size $2^i$ sampled in round $j$ and $\mathcal{D}_{ij}$ be the event that $S_{ij}$ has exactly one seed in $N_k(u_1) \cap N_k(u_2)$ and no other seeds in $N_k(u_1) \cup N_k(u_2)$. Let $\mathcal{D} = \cup_{j=1}^R \cup_{i=0}^r \mathcal{D}_{ij}$ and so $\mathcal{D}^c = \cap_{j=1}^R \cap_{i=0}^r \mathcal{D}_{ij}^c$. On the event $\mathcal{D}$, the seeds in the intersection will be one of the common seeds for computing the shortest distance according to Algorithm 1, and in that case, $\tilde{d}(u_1, u_2) \leq (1 + \varepsilon) \log_\lambda n$. Applying (van der Hofstad, 2024, Theorem 2.36), conditionally on $u_1, u_2$ to be in the same connected component, $d(u_1, u_2)/\log_\lambda n \xrightarrow{\mathbb{P}} 1$. Therefore, on $\mathcal{D}$, $\tilde{d}(u_1, u_2)$ provides a $(1 + \varepsilon)$-approximation of $d(u_1, u_2)$. Thus it suffices to show that $\lim_{n\to\infty} \mathbb{P}(\mathcal{D}) = 1$.

We will show that $\mathbb{P}(\mathcal{D}^c \mid G) \xrightarrow{\mathbb{P}} 0$, and consequently $\lim_{n\to\infty} \mathbb{P}(\mathcal{D}) = 1$ by the dominated convergence theorem. Since the choice of seeds in $S_i$'s are independent conditionally on $G$, with high probability,

$$
\mathbb{P}(\mathcal{D}^c \mid G) = \prod_{i=1}^R \prod_{i=0}^r \left( 1 - \frac{|N_k(u_1) \cap N_k(u_2)|}{n} \left( 1 - \frac{|N_k(u_1) \cup N_k(u_2)|}{n} \right)^{|S_{ij}|-1} \right)
$$

$$
\leq \exp\left( -R \sum_{i=0}^r \frac{|N_k(u_1) \cap N_k(u_2)|}{n} \left( 1 - \frac{|N_k(u_1) \cup N_k(u_2)|}{n} \right)^{2^i - 1} \right)
$$

$$
\leq \exp\left( -R \frac{|N_k(u_1) \cap N_k(u_2)|}{n} \sum_{i=0}^r \left( 1 - \frac{|N_k(u_1) \cup N_k(u_2)|}{n} \right)^{2^i} \right)
$$

$$
\leq \exp\left( -R \frac{|N_k(u_1) \cap N_k(u_2)|}{n} \sum_{i=0}^{r-1} \left( 1 - \frac{|N_k(u_1) \cup N_k(u_2)|}{n} \right)^{\frac{2^r}{r} i} \right)
$$

$$
\leq \exp\left( -R \frac{|N_k(u_1) \cap N_k(u_2)|}{n} \frac{1 - \left( 1 - \frac{|N_k(u_1) \cup N_k(u_2)|}{n} \right)^{2^r}}{1 - \left( 1 - \frac{|N_k(u_1) \cup N_k(u_2)|}{n} \right)^{\frac{2^r}{r}}} \right)
$$

$$
= \exp\left( -R n^{\varepsilon - 1 + o(1)} \frac{1 - \left( 1 - n^{\frac{\varepsilon}{2} - \frac{1}{2} + o(1)} \right)^{2^r}}{1 - \left( 1 - n^{\frac{\varepsilon}{2} - \frac{1}{2} + o(1)} \right)^{\frac{2^r}{r}}} \right)
$$

where the first "$\leq$" uses $1 - x \leq \exp(-x)$ for $x \geq 0$ and the second "$=$" follows from Proposition 3.5. Since $0 \leq \left( 1 - n^{\frac{\varepsilon}{2} - \frac{1}{2} + o(1)} \right)^{2^r} \leq \left( 1 - n^{\frac{\varepsilon}{2} - \frac{1}{2} + o(1)} \right)^{\frac{2^r}{r}} < \exp\left( -n^{\frac{\varepsilon}{2} - \frac{1}{2} + o(1)} \frac{1}{2} \frac{2^{\log n \frac{1-\varepsilon}{2\log 2}}}{\log n \frac{1-\varepsilon}{2\log 2}} \right) \to 0$ as $n \to \infty$, $R = \omega(n^{1-\varepsilon})$ is sufficient for the final bound to tend to 0.

## D  PROOF OF PROPOSITION 3.5

Fix $\varepsilon > 0$ (sufficiently small) and recall all the notation from Lemmas 3.3, 3.6. Let $\mathscr{F}_{k_1,k_2}$ be the minimum sigma-algebra with respect to which the random variables $(\partial N_j(u_1) : j \leq k_1)$, $(\partial N_j(u_2) : j \leq k_2)$ and the event $A_n$ are measurable. Let $\mathcal{E}_n$ be as defined in Lemma 3.3. Then, using Lemmas 3.3 and 3.6, we have $\lim_{n\to\infty} \mathbb{P}(\mathcal{E}_n \mid B_n) = \lim_{n\to\infty} \mathbb{P}(\mathcal{E}_n \mid A_n) = 1$. First, we prove the following: Fix any $\kappa_0 \log_\lambda n \leq k_1, k_2 \leq (\kappa + \kappa_0) \log_\lambda n$ such that $k_1 + k_2 \geq \log_\lambda n + 3\varepsilon$. Then, for all sufficiently large $n$,

$$
\mathbb{P}\left( |\partial N_{k_1}(u_1) \cap \partial N_{k_2}(u_2)| \in \left( n^{-2\varepsilon}(1 - \delta_n) \frac{\lambda^{k_1+k_2}}{n}, n^{2\varepsilon}(1 + \delta_n) \frac{\lambda^{k_1+k_2}}{n} \right) \,\middle|\, A_n \right) \geq 1 - n^{-\gamma_1}, \tag{12}
$$

for some $\gamma_1 = \gamma_1(\varepsilon) > 0$, $\gamma_2 = \gamma_2(\varepsilon) > 0$, and $\delta_n \leq n^{-c}$ for some $c > 0$. The choice of $\delta_n, \gamma_1, \gamma_2$ will become clear below. Let $I_{ij}$ be the indicator random variable for the edge $\{i, j\}$ being

present. Observe that $i \in \partial N_{k_1}(u_1) \cap \partial N_{k_2}(u_2)$ if and only if $i \in \partial N_{k_1}(u_1)$, $i \notin N_{k_2-1}(u_2)$ and there exists $j \in \partial N_{k_2-1}(u_2)$ such that $I_{ij} = 1$. Therefore, $|\partial N_{k_1}(u_1) \cap \partial N_{k_2}(u_2)| = \sum_{i \in \partial N_{k_1}(u_1) \setminus N_{k_2-1}(u_2)} \sum_{j \in \partial N_{k_2-1}(u_2)} I_{ij}$. Thus,

$$\mathbb{E}[|\partial N_{k_1}(u_1) \cap \partial N_{k_2}(u_2)| \mid \mathscr{F}_{k_1,k_2-1}]$$

$$= \left( |\partial N_{k_1}(u_1)| - \sum_{j \leq k_2-1} |\partial N_{k_1}(u_1) \cap \partial N_j(u_2)| \right) \times |\partial N_{k_2-1}(u_2)| \times \frac{\lambda}{n}. \tag{13}$$

On the event $\mathcal{E}_n$, $|\partial N_{k_1}(u_1)| \in (n^{-\varepsilon}\lambda^{k_1}, n^{\varepsilon}\lambda^{k_1})$ and $|\partial N_{k_2-1}(u_2)| \in (n^{-\varepsilon}\lambda^{k_2-1}, n^{\varepsilon}\lambda^{k_2-1})$, and by Lemma 3.3, $\mathbb{P}(\mathcal{E}_n \mid A_n) \geq 1 - n^{-\delta}$. Next, for any $j \leq k_2$, (13) yields that $\mathbb{E}[|\partial N_{k_1}(u_1) \cap \partial N_{j-1}(u_2)| \mid \mathscr{F}_{k_1,j-1}] \leq n^{\varepsilon}\lambda^{k_1+j}/n \leq \lambda^{k_1} n^{-\gamma_2}/(7k_2)$, where $\gamma_2 < 1 - \kappa - \kappa_0 - \varepsilon$ (note that $\gamma_2$ can be chosen to be positive for sufficiently small $\varepsilon$). Applying (Janson et al., 2000, Theorem 2.8 and Corollary 2.4), we have

$$\mathbb{P}(|\partial N_{k_1}(u_1) \cap \partial N_j(u_2)| > \lambda^{k_1} n^{-\gamma_2}/(7k_2) \mid \mathscr{F}_{k_1,j-1}) \leq \mathrm{e}^{-\lambda^{k_1} n^{-\gamma_2}/k_2} \leq \mathrm{e}^{-n^{\delta'}}, \tag{14}$$

for some $\delta' > 0$. Since the right hand side is deterministic function of $n$, the bound in (14) holds conditioned on $A_n$ as well. Thus, (13) yields, for all sufficiently large $n$, with probability at least $1 - n^{-\delta/2}$,

$$\mathbb{E}[|\partial N_{k_1}(u_1) \cap \partial N_{k_2}(u_2)| \mid \mathscr{F}_{k_1,k_2-1}] \in \left( \left(1 - \frac{\delta_n}{2}\right) n^{-2\varepsilon} \frac{\lambda^{k_1+k_2}}{n}, \left(1 + \frac{\delta_n}{2}\right) n^{2\varepsilon} \frac{\lambda^{k_1+k_2}}{n} \right),$$

where $\delta_n = o(n^{-\gamma_2})$.

When $k_1 + k_2 \geq \log_\lambda n + 3\varepsilon$, $\mathbb{E}[|\partial N_{k_1}(u_1) \cap \partial N_{k_2}(u_2)| \mid \mathscr{F}_{k_1,k_2-1}] \geq n^{\varepsilon/2}$. In that case, standard concentration inequalities for sums of independent Bernoulli random variables (Janson et al., 2000, Theorem 2.8 and Corollary 2.3, (2.9)) shows that $|\partial N_{k_1}(u_1) \cap \partial N_{k_2}(u_2)|$ concentrates around its expectation conditionally on $\mathscr{F}_{k_1,k_2-1}$, which proves (12).

Next, let $k_1, k_2$ be such that $k_1 + k_2 < \log_\lambda n + 3\varepsilon$. Then, (13) shows that, $\mathbb{E}[|\partial N_{k_1}(u_1) \cap \partial N_{k_2}(u_2)| \mid \mathscr{F}_{k_1,k_2-1}]\mathbb{1}_{\mathcal{E}_n} \leq n^{6\varepsilon}$ for all sufficiently large $n$. Again, an application of (Janson et al., 2000, Theorem 2.8 and Corollary 2.4) yields

$$\mathbb{P}(|\partial N_{k_1}(u_1) \cap \partial N_{k_2}(u_2)| > n^{7\varepsilon} \mid A_n) \leq \mathrm{e}^{-n^{7\varepsilon}} + n^{-\delta}. \tag{15}$$

Finally, combining (12) and (15), we conclude that, for all sufficiently large $n$, with probability at least $1 - 3(\log_\lambda n)^2 n^{-\min\{\gamma_1,\delta\}/3}$,

$$|N_k(u_1) \cap N_k(u_2)|$$

$$= \sum_{\substack{k_1,k_2 \leq k \\ k_1+k_2 \geq \log_\lambda n+3\varepsilon}} |\partial N_{k_1}(u_1) \cap \partial N_{k_2}(u_2)| + \sum_{\substack{k_1,k_2 \leq k \\ k_1+k_2 < \log_\lambda n+3\varepsilon}} |\partial N_{k_1}(u_1) \cap \partial N_{k_2}(u_2)| \tag{16}$$

$$\leq n^{3\varepsilon} \frac{\lambda^{2k}}{n} + n^{8\varepsilon} \leq n^{8\varepsilon}\left(\frac{\lambda^{2k}}{n} + 1\right),$$

and

$$|N_k(u_1) \cap N_k(u_2)| \geq \sum_{\substack{k_1,k_2 \leq k \\ k_1+k_2 \geq \log_\lambda n+3\varepsilon}} |\partial N_{k_1}(u_1) \cap \partial N_{k_2}(u_2)| \geq n^{-3\varepsilon}\frac{\lambda^{2k}}{n},$$

for all sufficiently large $n$. This concludes the proof for the asymptotics of $N_k(u_1) \cap N_k(u_2)$.

For part 2, note that $|N_k(u_i)| = \sum_{k_i \leq k} |\partial N_{k_i}(u_i)|$, and on the event $\mathcal{E}_n$, we have that $|\partial N_{k_i}(u_i)| \in (n^{-\varepsilon}\lambda^{k_i}, n^{\varepsilon}\lambda^{k_i})$ for all $k_i \leq k$ and $i = 1, 2$. Now, $\lambda^{2k}/n \leq \lambda^k n^{1-\kappa-\kappa_0}$ and $\kappa + \kappa_0 < 1$. Therefore, conditionally on $A_n$, with high probability,

$$|N_k(u_1) \cup N_k(u_2)| = |N_k(u_1)| + |N_k(u_2)| - |N_k(u_1) \cap N_k(u_2)| \in (n^{-2\varepsilon}\lambda^k, n^{2\varepsilon}\lambda^k).$$

Thus the proof follows.

# E  PROOF OF LEMMA 3.6

If $A_n$ occurs but $B_n$ does not then $|\mathscr{C}_{(2)}| \geq n^{\kappa_0-\varepsilon}$, which occurs with probability tending to zero, since $|\mathscr{C}_{(2)}| = O(\log n)$ w.h.p. On the other hand, if $B_n$ occurs and $A_n$ does not occur, then there exists $i$ such that either $|\partial N_L(u_i)| > n^{\kappa_0+\varepsilon}$ or $0 < |\partial N_L(u_i)| < n^{\kappa_0-\varepsilon}$. To bound the probabilities of these events, consider a branching process with progeny distribution being Poisson($\lambda$), and let $\mathcal{X}_l$ be the number of children at generation $l$. We first claim that, for any $\kappa_0 \in (0, \frac{1}{2})$ and $L = \kappa_0 \log_\lambda n$,

$$\lim_{n\to\infty} \mathbb{P}(|\partial N_L(u_i)| = \mathcal{X}_L) = 1. \tag{17}$$

Indeed, this is a consequence of (Bordenave, 2016, Lemma 3.13). Next, classical theory of branching processes shows that, on the event of survival, the growth rate of a branching process is exponential. More precisely, (Tanny, 1977, Theorem 5.5 (iii)) together with (Athreya and Ney, 1972, Theorem 2 on Page 8), it follows that

$$\lim_{L\to\infty} \mathbb{P}\big(L(1-\varepsilon) \leq \log_\lambda \mathcal{X}_L \leq L(1+\varepsilon), \mathcal{X}_L > 0\big) = 1$$

Therefore, $\lim_{L\to\infty} \mathbb{P}\big(n^{\kappa_0(1-\varepsilon)} \leq \mathcal{X}_L \leq n^{\kappa_0(1+\varepsilon)}, \mathcal{X}_L > 0\big) = 1$. Since $\kappa_0 - \varepsilon < \kappa_0(1-\varepsilon)$ and $\kappa_0 + \varepsilon > \kappa_0(1+\varepsilon)$,

$$\lim_{L\to\infty} \mathbb{P}\big(n^{\kappa_0-\varepsilon} \leq \mathcal{X}_L \leq n^{\kappa_0+\varepsilon}, \mathcal{X}_L > 0\big) = 1 \tag{18}$$

Combining (17) and (18), it follows that

$$\mathbb{P}(B_n \setminus A_n) \leq \sum_{i=1,2} \mathbb{P}(0 < |\partial N_L(u_i)| < n^{\kappa_0-\varepsilon} \text{ or } |\partial N_L(u_i)| > n^{\kappa_0+\varepsilon}) \to 0.$$

# F  EXPERIMENT DETAILS

In our experiments, we train GNNs to learn to compute the shortest path distances from every seed to every node in sparse, undirected, and unweighted connected random graphs. Using the trained GNNs, we generate node embeddings as in local step of Algorithm 1. Finally, we evaluate the performance of the embeddings in shortest path approximations and test the model's transferability.

To construct the GNNs, we consider four standard GNN architectures (GCN, GraphSage, GAT, and GIN) with sum aggregation, dropout and ReLU between the convolutions, and ReLU activation function. For each GNN architecture, we experiment with nine models that differ in widths and depths of their hidden layers. The first and last GNN layers both consist of $\lfloor \sqrt{n} \rfloor$ nodes, which correspond to $\lfloor \sqrt{n} \rfloor$ seeds inputted into the GNNs. The widths and depths of the hidden layers are as follows:

- Depth-6 GNNs: 128-64-32-16, 64-32-16-8, 32-16-8-4
- Depth-5 GNNs: 128-64-32, 64-32-16, 32-16-8
- Depth-4 GNNs: 128-64, 64-32, 32-16

We train our GNNs on ER graphs generated by $ER_n(\lambda/n)$. To ensure that the graph are sparse and each has a giant component with high probability, it is necessary to have $1 < \lambda \ll n$. We thus evaluate $\lambda \in \{3, 4, 5, 6\}$ with $n \in \{25, 50, 100, 200, 400, 800, 1600, 3200\}$. We treat each graph as a batch of nodes and have train-validation-test size of 200-50-50 batches. The training occurs in 1000 epochs with early stopping patience of 100 epochs, mean squared error (MSE) loss, Adam optimizer with a learning rate of 0.01 and weight decay of 0.0001, and a cyclic-cosine learning rate scheduler with cyclical learning rate between 0.001 and 0.1 for 10 iterations in the increasing half in combination with the default cosine annealing learning rate for a maximum of 20 iterations.

All experiments were run using PyTorch Geometric Fey and Lenssen (2019) on a Lambda Vector 1 machine with an AMD Ryzen Threadripper PRO 5955WX CPU (16 cores), 128 GB RAM, and two NVIDIA GeForce RTX 4090 GPUs (without parallel training).

The code can be found at `https://github.com/ruiz-lab/shortest-path`.

## G   MORE EXPERIMENTAL RESULTS

We present additional experimental results that provide deeper insights into the GNNs and the GNN-augmented algorithm for computing shortest distances.

### G.1   EXPERIMENT 1

We consider the graph model $ER_n(\lambda/n)$ which generates graphs that are more likely to be less sparse ($\lambda = 3$) or more sparse ($\lambda = 6$) than those described in Section 4. Figure 5 shows that, using the same GNN, the prediction curve remains consistent over the same actual distance range. Once the GNN predictions enter the saturated region, they remain saturated even for larger actual distances in less sparse graphs.

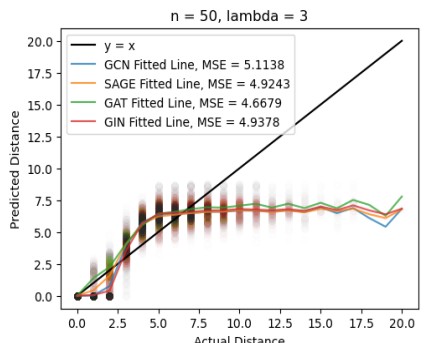 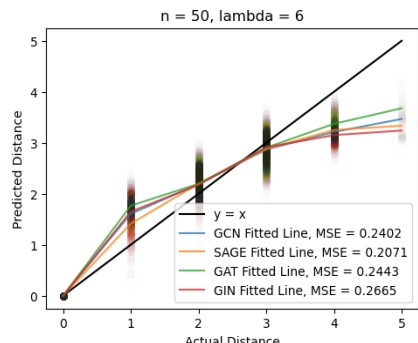

Figure 5: Raw outputs of $\lfloor \sqrt{n} \rfloor$-64-32-16-$\lfloor \sqrt{n} \rfloor$ GNNs that are trained on ER graphs generated by $ER_n(\lambda/n)$ with $\lambda \in \{3, 6\}$ to predict shortest path distances end-to-end. Evaluation data are ER graphs from the same model.

### G.2   EXPERIMENT 2

As seen earlier, the GNN prediction curve is similar for $\lambda = 3$ and $\lambda = 4$ under the graph model $ER_n(\lambda/n)$, with more distances in graphs from $ER_n(3/n)$ falling into the saturated region than those in graphs from $ER_n(4/n)$ (as they are likely more sparse).

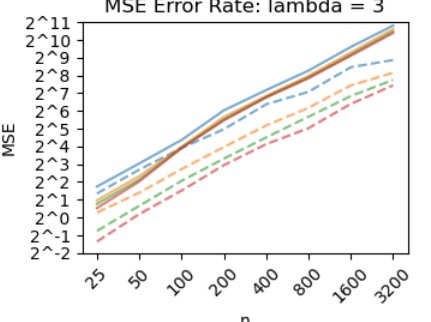 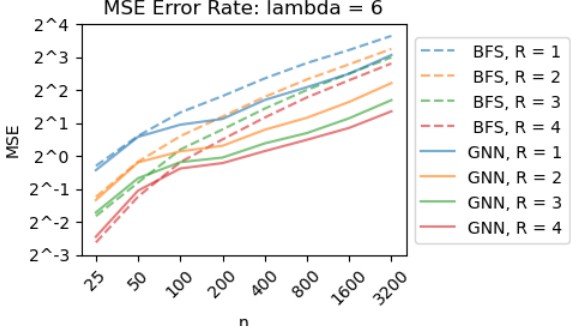

Figure 6: Performance of BFS-based embeddings vs. GNN-based embeddings with GNNs trained on ER graphs generated by $ER_n(\lambda/n)$ for $\lambda \in \{3, 6\}$.

Since GNN-based embeddings on graphs from $ER_n(4/n)$ were not as effective as BFS-based embeddings in estimating shortest-path distances with the local-global algorithm, it is not surprising that GNN-based embeddings on graphs from $ER_n(3/n)$ also perform worse than BFS-based embeddings in terms of MSE, as shown in Figure 6. On the other hand, since graphs from $ER_n(6/n)$ are more likely to have distances falling in the predictable region of the GNN compared to graphs from

$ER_n(5/n)$, and the GNN-based embeddings on graphs from $ER_n(5/n)$ perform better than BFS-based embeddings in estimating shortest-path distances with the local-global algorithm, GNN-based embeddings on graphs from $ER_n(6/n)$ also result in lower MSE than BFS-based embeddings.

## G.3 EXPERIMENT 3

We repeat Experiment 3 in Section 4 with $\lambda = 6$, where $ER_n(\lambda/n)$ generates graphs that are more likely to be less sparse. As seen from Figure 7, the resulting MSE curves are consistent with those in Section 4 that the MSE decreases as the training graph size increases and GNNs outperform BFS when the training graph size exceeds 100.

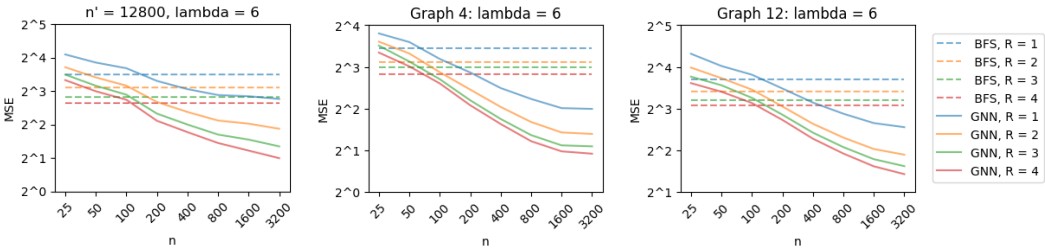

Figure 7: Error rates on test ER graphs generated by $ER_{n'}(\lambda/n')$ (a), a GEMSEC social network with 14,113 nodes (b), and a Arxiv collaboration network with 28,281 nodes (c) by BFS-based embeddings vs. GNN-based embeddings using GNNs trained on ER graphs from $ER_n(\lambda/n)$ with $\lambda = 6$.

Finally, we present additional transferability results of the local-global algorithm using BFS-based and GNN-based embeddings on a larger set of real benchmark graphs, where the GNNs are trained on $ER_n(\lambda/n)$ with $\lambda = 5$.

Table 1: Details on the largest connected component of selected real networks.

|  | Name | Category | # of Nodes | # of Edges |
|---|---|---|---|---|
| 1 | GEMSEC Athletes Rozemberczki et al. (2019b) | Social Network | 13,866 | 86,858 |
| 2 | GEMSEC Public Figures Rozemberczki et al. (2019b) | Social Network | 11,565 | 67,114 |
| 3 | GEMSEC Politicians Rozemberczki et al. (2019b) | Social Network | 5,908 | 41,729 |
| 4 | GEMSEC Companies Rozemberczki et al. (2019b) | Social Network | 14,113 | 52,310 |
| 5 | GEMSEC TV Shows Rozemberczki et al. (2019b) | Social Network | 3,892 | 17,262 |
| 6 | Twitch-EN Rozemberczki et al. (2019a) | Social Network | 7,126 | 35,324 |
| 7 | Deezer EuropeRozemberczki and Sarkar (2020) | Social Network | 28,281 | 92,752 |
| 8 | LastFM AsiaRozemberczki and Sarkar (2020) | Social Network | 7,624 | 27,806 |
| 9 | Arxiv COND-MAT Leskovec et al. (2007) | Collaboration Network | 21,364 | 91,315 |
| 10 | Arxiv GR-QC Leskovec et al. (2007) | Collaboration Network | 4,158 | 13,425 |
| 11 | Arxiv HEP-PH Leskovec et al. (2007) | Collaboration Network | 11,204 | 117,634 |
| 12 | Arxiv HEP-TH Leskovec et al. (2007) | Collaboration Network | 8,638 | 24,817 |
| 13 | Oregon Autonomous System 1 Leskovec et al. (2005) | Autonomous System | 11,174 | 23,409 |
| 14 | Oregon Autonomous System 2 Leskovec et al. (2005) | Autonomous System | 11,461 | 32,730 |
| 15 | GEMSEC Artists Rozemberczki et al. (2019b) | Social Network | 41,618 | 557,133 |
| 16 | ER-AVGDEG10-100K-L2 Rossi and Ahmed (2015) | Labeled Network | 99,997 | 499,359 |
| 17 | Brightkite Rossi and Ahmed (2015) | Social Network | 56,739 | 212,945 |

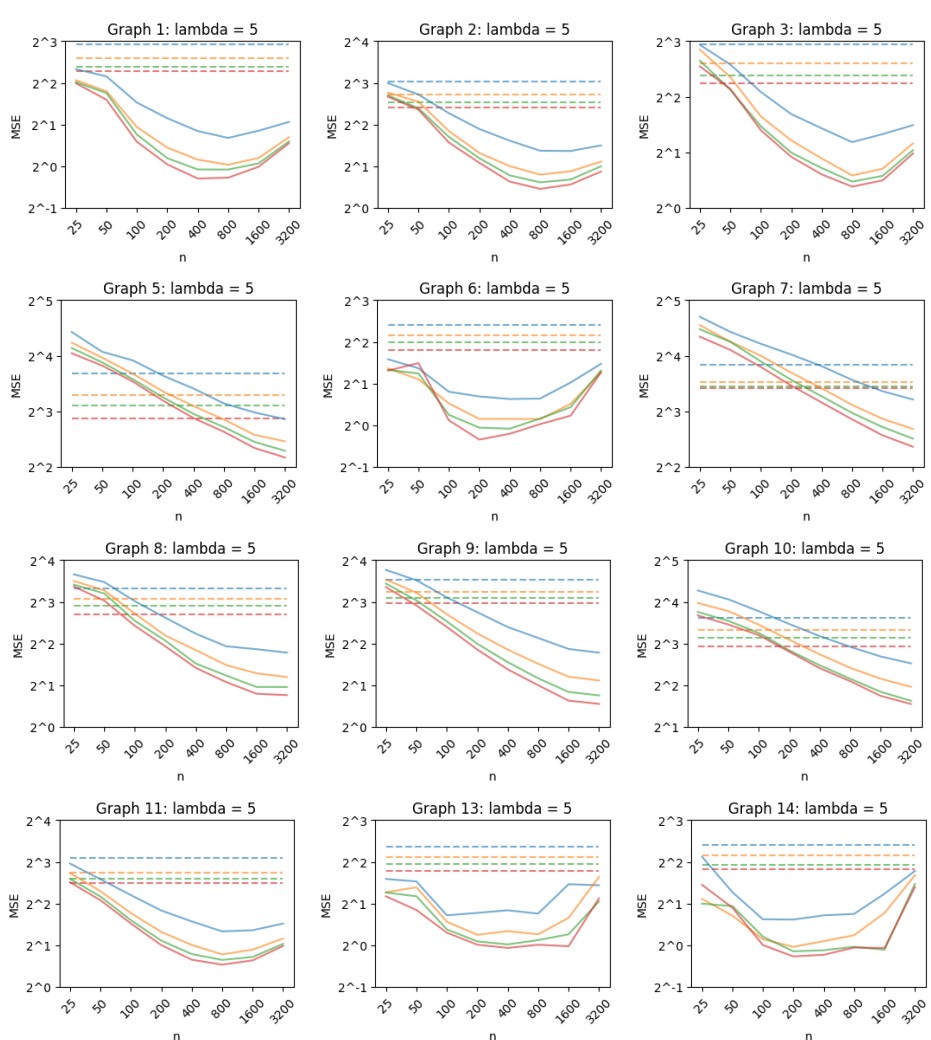

Figure 8: Additional transferability results on real networks by BFS-based embeddings vs. GNN-based embeddings using GNNs trained on ER graphs generated by $ER_n(\lambda/n)$ with $\lambda = 5$. Legend is the same as in Figure 7.

