# OpenReview forum: "Local-Global Shortest Path Algorithms on Random Graphs, Enhanced with GNNs"
_ICLR.cc/2025/Conference — Submitted to ICLR 2025_

### Official Review · Reviewer_FQhS · 2024-10-31

**Soundness:** 2
**Presentation:** 2
**Contribution:** 1
**Rating:** 1
**Confidence:** 4

**Summary:**

This paper addresses the shortest-path distance problem using a machine learning approach. The authors employ a local-global algorithm to approximate shortest-path distances, analyzing the lower and upper bound distortions of these approximations on Erdős–Rényi (ER) random graphs. They propose using a GNN to implement the local step of the algorithm. Experimental results are reported on ER graphs and two social network datasets.

**Strengths:**

**Originality**

The concept of the local-global algorithm is largely drawn from existing work, and using GNNs to implement the local step does not introduce much new insights. The main contribution of this paper lies in the analysis of lower and upper bound distortions on Erdős–Rényi random graphs.

**Quality**

I have concerns about the proposed method, as it appears to be effective only on small graphs, even when computing approximate distances. This limitation raises questions about its scalability and practical applicability to larger networks.

**Clarity**

The presentation could benefit from improvements in several areas. For instance, the research problem (tackling the shortest-path distance problem using machine learning) lacks clear motivation. The transferability of the GNN from small graphs to large graphs is not adequately justified. The experiments are limited, and the results are not thoroughly explained.


**Significance**

The paper overlooks a substantial body of literature (e.g., [1,2,3,4]) on fast and scalable shortest-path algorithms that can compute exact shortest-path distances on large networks (with millions or even billions of nodes) within milliseconds per query, and in some cases, even microseconds for specific network types like road networks. Given these existing solutions, the significance of this work is questionable.

References:

[1] Fast Exact Shortest-Path Distance Queries on Large Networks by Pruned Landmark Labeling, Akiba et al. SIGMOD 2012

[2] Fully Dynamic Shortest-Path Distance Query Acceleration on Massive Networks, Hayashi et al. CIKM 2016

[3] A Highly Scalable Labelling Approach for Exact Distance Queries in Complex Networks, Farhan, et al., EDBT 2019

[4] When Hierarchy Meets 2-Hop-Labeling: Efficient Shortest Distance Queries on Road Networks, Quyang, SIGMOD 2018

**Weaknesses:**

W1: The research problem may need to be revisited or reconsidered. The authors argue that “modern networks often consist of billions of nodes, and global algorithms take hours to be implemented, whereas approximate solutions are typically needed with ultra-low latency.” However, this claim lacks citations or sources of information. Can the authors provide specific citations supporting the claim about computational challenges on very large networks? In recent literature on shortest-path distance algorithms, for complex networks with billions of nodes, it often takes only a couple of hours to build an index, with exact shortest-path distance queries completing in under one microsecond [1].  To better justify the need for an approximate method, I would suggest a comparison with existing efficient exact algorithms.


W2: The proposed method is primarily analyzed and evaluated based on Erdős–Rényi (ER) random graphs. However, it is well known that shortest-path distance problems on different graph structures may require different algorithmic designs. For instance, road networks typically have low node degrees, while social networks often have a dense core, leading to significantly different design choices. Therefore, the results on ER random graphs have limited applicability in practice.

W3: The design of the GNN appears to follow the basic principles of message passing within a local neighborhood. However, it is unclear how long-distances would be managed if the GNN depth is small. Conversely, increasing the GNN depth could lead to the oversmoothing problem. Also, since the shortest-path distance problem is to calculate the number of edges in the shortest path between two vertices, it is unclear how the node features contribute meaningfully in this context. Providing examples or intuitive explanations would help clarify this aspect. Finally, the rationale behind the transferability of the model is not well-justified in this setting and would benefit from further clarification.

W4: The experiments do not align with the stated motivation of the work, which criticizes traditional approaches for the high computational cost of shortest-path calculations. Most experiments are conducted on Erdős–Rényi graphs with sizes ranging from just 25 nodes to 3200 nodes, which fails to demonstrate the scalability of the proposed method on large networks. In Figure 2, even on a small graph with 50 nodes, the GNN performs poorly in predicting distances when the distances are larger, which raises concerns about its effectiveness. In Figures 4(b)–(c), why the MSE of BFS remains flat as n increases from 25 to 3200, whereas in Figure 3, the MSE of BFS increases with n?

There are various real-world graphs and networks commonly used by researchers to benchmark shortest-path distance algorithms (e.g., [1,2,3,4]), such as road networks with sizes up to 24 million nodes and complex networks (including social networks) ranging from 1 million to 2 billion nodes. However, the paper only benchmarks on small ER graphs and two small real world social networks against the BFS algorithm. It would be more informative to compare the proposed method with state-of-the-art shortest-path algorithms on these larger, real-world datasets to better assess its effectiveness and scalability.

**Questions:**

See the questions in "Weaknesses"

---

> ### Author Response · Authors · 2024-11-20
> **Thank you for your review. Your comments are addressed point-by-point below.**
>
> > **I have concerns about the proposed method, as it appears to be effective only on small graphs, even when computing approximate distances. This limitation raises questions about its scalability and practical applicability to larger networks**
>
> We thank the reviewer for the useful feedback.
>
> Note that we report results on larger networks, such as in Figures 4a, where the target graphs are ER graphs with as many as 12,800 nodes and 38,400 edges, and in Figures 4b and 4c, where the target graphs are real-world networks with as many as 28,281 nodes and 92,752 edges.
>
> We do however understand the reviewer's criticism that the networks in our experiments are not sufficiently large, and agree that numerical results on larger graphs would be more compelling. For that reason, we plan to run transferability experiments on SNAP's Amazon co-purchasing network, with around 100k nodes, and Youtube network, with around 1 million nodes. These results will be included in Section 4.3 and reported as they become available.
>
> The lack of experiments on larger networks in the original submission was primarily due to limited computational resources. Further, our paper's contribution is primarily theoretical---more specifically an analysis of approximation factors/embedding sizes for local-global algorithms on ER graphs. The addition of GNNs in the global step is a methodological contribution inspired by the fact that, on ER graphs, which can be thought of as being the average-case as opposed to the worst-case graphs in Das Sarma et al. (2010) and Bourgain (1985), the required embedding size (and thus the required GNN depth) are lower.
>
> > **The presentation could benefit from improvements in several areas. For instance, the research problem [...] lacks clear motivation. The transferability of the GNN from small graphs to large graphs is not adequately justified. The experiments are limited [...].**
>
> Thank you for bringing this up.
>
> **Motivation and contributions.** We will improve the writing to address this.
> The main contribution of our manuscript is an analysis of local-global shortest path (SP) algorithms on ER graphs.
> The idea was to initiate the study of average case analysis in the same spirit as the classical works in the theoretical CS literature for NP-hard problems.
> Our goal is slightly different though, since the underlying problem is not NP-hard, and the average case analysis is used for understanding possible reduction in complexity. We will add a few lines along this direction in the introduction to improve clarity in our stated motivation.
>
> **Transferability justification.** The reason for leveraging GNN transferability has to do with the fact that the cost of training GNNs increases with the graph size. But indeed, it seems counterintuitive to use this property in shortest path computations since, as the graph size $n$ increases, the average path length increases as $\log (n)$ while the GNN depth remains fixed. To curb this limitation, we use the same GNN but increase the number of seed sets by running the GNN inference step $\sqrt{N}/\sqrt{n}$ times, where $n$ is the size of the smaller training graph and $N$ is the size of the large target graph. This corresponds to $\sqrt{N}/\sqrt{n}$ different random seed sets, whose embeddings are then appended to form a $\sqrt{N}$-dimensional embedding per node. We will better emphasize this step in the paper.
>
> **Experimental results.** In addition to the experiments on synthetic ER graphs with up to 12,800 nodes and 38,400 edges, we have also reported results on real-world social networks with as many as 28,281 nodes and 92,752 edges. We could not fit all of these results in the main body due to the page limit, but the complete set of experiments is reported in the appendices.
>
> We acknowledge that the graph sizes in our experiments are not as large, and we thank the reviewer for pointing that out. We will make our best effort to overcome this limitation in the coming week, resource-permitting; but we respectfully point out that this paper's contribution as primarily theoretical and, as such, the main goal of the numericals is illustrating the empirical manifestation of our theoretical findings.

---

> > ### Author Response · Authors · 2024-11-20
> > **Part 2**
> >
> > > **The paper overlooks a substantial body of literature (e.g., [1,2,3,4]) on fast and scalable shortest-path algorithms that can compute exact shortest-path distances on large networks (with millions or even billions of nodes) within milliseconds per query, and in some cases, even microseconds for specific network types like road networks. Given these existing solutions, the significance of this work is questionable.**
> >
> > >**References:**
> >
> > >**[1] Fast Exact Shortest-Path Distance Queries on Large Networks by Pruned Landmark Labeling, Akiba et al. SIGMOD 2012**
> >
> > >**[2] Fully Dynamic Shortest-Path Distance Query Acceleration on Massive Networks, Hayashi et al. CIKM 2016**
> >
> > >**[3] A Highly Scalable Labelling Approach for Exact Distance Queries in Complex Networks, Farhan, et al., EDBT 2019**
> >
> > >**[4] When Hierarchy Meets 2-Hop-Labeling: Efficient Shortest Distance Queries on Road Networks, Quyang, SIGMOD 2018**
> >
> > Thank you for these references. We agree with the reviewer that shortest path finding is a fundamental problem and speeding up queries a substantial challenge. We acknowledge the large body of work focusing on the shortest path problem in the applied community, and agree that important references from the area were missing from our paper. We will cite these and other relevant works in the revision. Once again, thank you for pointing to these references and helping us improve our paper.
> >
> > We would like to however clarify that the primary goal of our paper was not to introduce a new state-of-the-art method for SP calculations, but rather to provide a theoretical analysis of existing local-global SP algorithms on ER graphs.
> > The idea behind this analysis is to establish guarantees for the approximation factor/embedding size of these algorithms that are more akin to average-case guarantees commonly studied in theoretical computer science, as existing theoretical analyses of local-global SP algorithms have only focused on worst-case guarantees.
> >
> > > **The research problem may need to be revisited or reconsidered. The authors argue that “modern networks often consist of billions of nodes, and global algorithms take hours to be implemented, whereas approximate solutions are typically needed with ultra-low latency.” However, this claim lacks citations or sources of information. Can the authors provide specific citations supporting the claim about computational challenges on very large networks? In recent literature on shortest-path distance algorithms, for complex networks with billions of nodes, it often takes only a couple of hours to build an index, with exact shortest-path distance queries completing in under one microsecond [1]. To better justify the need for an approximate method, I would suggest a comparison with existing efficient exact algorithms.**
> >
> > Thanks for raising these concerns.
> >
> > **Reference for our statement.** What we meant by this sentence is that by using classical global algorithms such as Djikstra's, it could take hours or days to find all shortest paths on billion-node graphs. This is a common argument in relevant papers on the SP problem, see, e.g., Das Sarma et al. (2012) and Awasthi et al. (2022). We also note that the inclusion of this statement was meant to motivate the importance and the difficulty of the shortest path problem, but it is not our research question. The primary research question of our paper was the relationship between the approximation factor and the required embedding size of local-global SP algorithms on ER graphs.
> >
> > Still, we agree that in the current form the statement is very broad.
> > We will make it more precise and provide references mentioned above.
> >
> > **Relevance of approximate methods.** Approximate methods can be particularly useful in scenarios where resources may not be sufficient to compute a full index or where memory limitations make storing such an index challenging. The approximate methods we consider come with guarantees, which help to quantify potential deviations from exact solutions. Learning-enhanced approximate methods, such as the GNN-based algorithm we experiment with, give additional flexibility: once the ML model has been trained for sketch computation, inference is fast and inexpensive, allowing for sketches to be computed online. This reduces the need for precomputing and storing large-scale sketches, which could be beneficial in memory-scarce scenarios.

---

> > > ### Author Response · Authors · 2024-11-20
> > > **Part 3**
> > >
> > > > **The proposed method is primarily analyzed and evaluated based on Erdős–Rényi (ER) random graphs. However, it is well known that shortest-path distance problems on different graph structures may require different algorithmic designs. For instance, road networks typically have low node degrees, while social networks often have a dense core, leading to significantly different design choices. Therefore, the results on ER random graphs have limited applicability in practice.**
> > >
> > > This is a good point. The reason for choosing ER graphs is that starting with simpler models like ER graphs enables us to develop the theoretical tools and insights necessary for analyzing algorithms on more complex graph structures.
> > > We believe that the techniques presented in this paper can be extended to graphs with good expansion properties and more realistic models, such as inhomogeneous random graphs and configuration models, which are commonly used to model social networks. However, analyzing these models is expected to introduce additional complexities due to inhomogeneous connectivity patterns and dependencies among edge occupancy variables. For this reason, we have left such extensions as future work.
> > >
> > > For road networks, which are typically planar, we believe that different techniques will be required to achieve comparable results.
> > >
> > > > **The design of the GNN appears to follow the basic principles of message passing within a local neighborhood. However, it is unclear how long-distances would be managed if the GNN depth is small. Conversely, increasing the GNN depth could lead to the oversmoothing problem.**
> > >
> > > The depth of a GNN is indeed an important consideration. As shown by Loukas (2020), GNNs inherit provable limitations when applied to the shortest path problem. Local-global algorithms aim to overcome these limitations. For ER graphs, these algorithms intuitively require reduced depth because ER graphs have good expansion properties, enabling them to establish short connections with the seed sets.
> > >
> > > Regarding oversmoothing, we specifically keep the GNN depth small to minimize its impact. Oversmoothing typically arises in deeper GNNs where repeated aggregation causes node representations to converge, but this is unlikely to be observed with the depths used in our setup.
> > >
> > > > **Also, since the shortest-path distance problem is to calculate the number of edges in the shortest path between two vertices, it is unclear how the node features contribute meaningfully in this context. Providing examples or intuitive explanations would help clarify this aspect.**
> > >
> > > We appreciate this question and understand the need for clarification. The use of node embeddings to compute shortest-path distances is a standard approach, even in the original local-global algorithms by Bourgain and Das Sarma et al. The key idea is to embed the graph into a Hilbert space in a way that approximately preserves the pairwise distances.
> > > Our approach follows this well-established methodology, leveraging node features as part of the embedding process.
> > >
> > > > **Finally, the rationale behind the transferability of the model is not well-justified in this setting and would benefit from further clarification.**
> > >
> > > The reason for leveraging GNN transferability has to do with the fact that the cost of training GNNs increases with the graph size. But indeed, it seems counterintuitive to use this property in shortest path computations since, as the graph size $n$ increases, the average path length increases as $\log (n)$ while the GNN depth remains fixed. To curb this limitation, we use the same GNN but increase the number of seed sets by running the GNN inference step $\sqrt{N}/\sqrt{n}$ times, where $n$ is the size of the smaller training graph and $N$ is the size of the large target graph. This corresponds to $\sqrt{N}/\sqrt{n}$ different random seed sets, whose embeddings are then appended to form a $\sqrt{N}$-dimensional embedding per node. We will better emphasize this step in the paper.

---

> > > > ### Author Response · Authors · 2024-11-20
> > > > **Part 4**
> > > >
> > > > > **The experiments do not align with the stated motivation of the work, which criticizes traditional approaches for the high computational cost of shortest-path calculations. Most experiments are conducted on Erdős–Rényi graphs with sizes ranging from just 25 nodes to 3200 nodes, which fails to demonstrate the scalability of the proposed method on large networks.**
> > > >
> > > > We appreciate the reviewer's concern. While we do report results on larger networks in Figure 4, including ER graphs with up to 12,800 nodes and 38,400 edges and real-world networks with up to 28,281 nodes and 92,752 edges, we understand that these may still not be sufficiently large to demonstrate full scalability. We agree that experiments on larger graphs would be more compelling and are currently running transferability experiments on SNAP's Amazon co-purchasing (around 100k nodes) and Youtube networks (around 1 million nodes), which we will report as they become available in the coming week.
> > > >
> > > > Regarding the stated motivation, we did not intend to criticize traditional approaches or suggest that our method is designed to empirically outperform them. Our mention of classical global algorithms such as Dijkstra's taking a long time on very large graphs was intended to highlight the inherent difficulty and importance of the shortest-path problem. This argument is common in the relevant literature, including works like Das Sarma et al. (2012) and Awasthi et al. (2022). The primary research question of our paper was the relationship between the approximation factor and the required embedding size of local-global SP algorithms on ER graphs.
> > > >
> > > > >**In Figure 2, even on a small graph with 50 nodes, the GNN performs poorly in predicting distances when the distances are larger, which raises concerns about its effectiveness. In Figures 4(b)–(c), why the MSE of BFS remains flat as n increases from 25 to 3200, whereas in Figure 3, the MSE of BFS increases with n?**
> > > >
> > > > **Figure 2.** The purpose of Figure 2 is to demonstrate that using GNNs for end-to-end shortest-path computations is not effective. This result aligns with our argument that GNNs alone are insufficient for this task.
> > > >
> > > > **Figures 4b and 4c.** In Figure 4, we compare the MSE of the GNN-enhanced local-global algorithm with the original Bourgain algorithm. The GNNs are trained on graphs of increasing size (x-axis), while the target graphs remain large networks with 12.8k (Fig. 4a), 14k (Fig. 4b), and 29k (Fig. 4c) nodes, respectively. For the original Bourgain algorithm, the BFS sketches are computed directly on these large target graphs, which is why its MSE remains flat across graph sizes.
> > > >
> > > > **Figure 3.** In this figure, we report the MSE for both the GNN-enhanced and BFS-based local-global methods on graphs of increasing size (x-axis). In this experiment, there is no transferability—sketches are computed on the same graphs for which performance is evaluated.
> > > >
> > > > > **There are various real-world graphs and networks commonly used by researchers to benchmark shortest-path distance algorithms (e.g., [1,2,3,4]), such as road networks with sizes up to 24 million nodes and complex networks (including social networks) ranging from 1 million to 2 billion nodes. However, the paper only benchmarks on small ER graphs and two small real world social networks against the BFS algorithm. It would be more informative to compare the proposed method with state-of-the-art shortest-path algorithms on these larger, real-world datasets to better assess its effectiveness and scalability.**
> > > >
> > > > We agree that experiments on larger graphs would strengthen the paper and are actively running transferability experiments on SNAP's Amazon co-purchasing (100k nodes) and Youtube networks (1 million nodes), which we will report as they become available in the coming week.
> > > >
> > > > Regarding comparisons with state-of-the-art shortest-path (SP) algorithms, we respectfully point out that that is beyond the scope of our paper. The primary objective of our work is not to propose a new state-of-the-art method for SP calculations but to analyze local-global SP algorithms on ER graphs. Specifically, our focus is on establishing guarantees for the approximation factor and embedding size of these algorithms, drawing parallels to average-case guarantees commonly studied in theoretical computer science. This is in contrast to existing analyses, which primarily emphasize worst-case guarantees.

---

> > > > > ### Comment · Reviewer_FQhS · 2024-11-24
> > > > > **Official Comment by Reviewer FQhS**
> > > > >
> > > > > Thank you for the rebuttal. I appreciate the authors' efforts in addressing my concerns. However, my main concerns remain unresolved. Below, I outline some of them in detail.
> > > > >
> > > > > > "Motivation and contributions. We will improve the writing to address this...."
> > > > >
> > > > > Thank you for committing to making these changes later. However, I’m unclear why these updates cannot be incorporated into the current revised version. Is there a specific reason preventing this?
> > > > >
> > > > > > "We also note that the inclusion of this statement was meant to motivate the importance and the difficulty of the shortest path problem, but it is not our research question."
> > > > >
> > > > > This statement is inaccurate and does not reflect the latest research progress in the field. Including it in the paper, even as part of the motivation, is misleading. The current version should be revised to ensure accuracy and clarity.
> > > > >
> > > > > >"Approximate methods can be particularly useful in scenarios where resources may not be sufficient to compute a full index or where memory limitations make storing such an index challenging."
> > > > >
> > > > > Given current research progress, these scenarios do not apply to shortest path distance methods, as distance information can be efficiently stored in an index with reasonable space requirements. However, the situation may differ for shortest path algorithms, which could encounter such scenarios. This paper focuses specifically on shortest path distance algorithms, not shortest path algorithms.
> > > > >
> > > > > >"Regarding oversmoothing, we specifically keep the GNN depth small to minimize its impact. ..."
> > > > >
> > > > > I am still unclear why the GNNs presented in Eq. (7), as well as the GNN models used in the experiments (GCN, GraphSAGE, GAT, and GIN), are capable of capturing shortest path distances on graphs, especially when the GNN depth is small. To clarify, please provide a formal proof or theoretical justification to how shortest path distances can be captured by the proposed GNN architecture.
> > > > >
> > > > > >"... Our approach follows this well-established methodology, leveraging node features as part of the embedding process."
> > > > >
> > > > > This doesn't answer my question on " it is unclear how the node features contribute meaningfully in this context. Providing examples or intuitive explanations would help clarify this aspect."
> > > > >
> > > > > The reason for this request is that, unlike many other graph properties, shortest path distances depend solely on graph structures and are independent of node features. Therefore, it is unclear what kind of node features was used in the experiments where GNNs were applied for local steps. Please include a concrete example to clarify this.
> > > > >
> > > > > >"The reason for leveraging GNN transferability has to do with the fact that the cost of training GNNs increases with the graph size..."
> > > > >
> > > > > I fail to see how this approach ensures GNN transferability. Unlike other graph properties, distance information does not appear to adhere to consistent patterns, as the same node can have varying distances to different nodes.

---

> > > > > > ### Author Response · Authors · 2024-11-25
> > > > > > **Thank you for your response to our rebuttal**
> > > > > >
> > > > > > Thanks for your response to our rebuttal! We have just uploaded a preliminary revision of our paper for your reference. We address your other concerns point-by-point below.
> > > > > >
> > > > > > > **This statement is inaccurate and does not reflect the latest research progress in the field. Including it in the paper, even as part of the motivation, is misleading. The current version should be revised to ensure accuracy and clarity.**
> > > > > >
> > > > > > We have softened this statement in the revision. We now say:
> > > > > >
> > > > > > ``Finding shortest paths on networks is an important combinatorial optimization problem arising in many practical applications, such as transportation networks (Fu et al., 2006) and integrated circuit design (Cong et al., 1998). Unlike other optimization problems on graphs, exact solutions for shortest paths can be found using classical algorithms such as Dijkstra's algorithm in polynomial time. Moreover, advancements in indexing techniques have made exact shortest-path distance queries highly efficient, with solutions capable of handling large-scale graphs and providing microsecond-level query times in certain settings [1--4 in your review above].''
> > > > > >
> > > > > > We hope these changes address your concerns.
> > > > > >
> > > > > > > **Given current research progress, these scenarios do not apply to shortest path distance methods, as distance information can be efficiently stored in an index with reasonable space requirements. However, the situation may differ for shortest path algorithms, which could encounter such scenarios. This paper focuses specifically on shortest path distance algorithms, not shortest path algorithms.**
> > > > > >
> > > > > > We acknowledge the reviewer’s observation that the scenarios described may not apply to shortest path distance methods under current progress. We thank the reviewer for raising this important point and have revised the text to ensure clear differentiation between shortest path algorithms and shortest path distance methods.
> > > > > >
> > > > > > We also thank the reviewer for pointing out that distance indexing methods can efficiently store distance information for many scenarios. However, our motivation stems from scenarios where such indexes are impractical---for example, in dynamic networks where distances need to be frequently updated, or in applications with strict memory or real-time inference constraints. We have revised the text to clarify these distinctions as follows:
> > > > > >
> > > > > > ``However, not all scenarios allow for such efficient indexing. For example, dynamic networks with frequently updated edge weights or applications requiring real-time computation on resource-constrained devices may not benefit from precomputed indexes. In such cases, approximate methods are particularly valuable due to their adaptability and lower computational overhead. This has motivated the exploration of machine learning approaches to shortest path finding, particularly those employing graph neural networks (GNNs).''
> > > > > >
> > > > > > Finally, we once again stress that our work seeks to analyze local-global methods' theoretical properties in ER random graph settings, attempting at something close to an average-case guarantee. We do believe such an analysis is valuable, given its absence in the current literature and the relevance of approximation algorithms based on Bourgain's seminal result. We have revised the text to emphasize that.
> > > > > >
> > > > > > > **I am still unclear why the GNNs presented in Eq. (7), as well as the GNN models used in the experiments (GCN, GraphSAGE, GAT, and GIN), are capable of capturing shortest path distances on graphs, especially when the GNN depth is small. To clarify, please provide a formal proof or theoretical justification to how shortest path distances can be captured by the proposed GNN architecture.**
> > > > > >
> > > > > > We thank the reviewer for this question. The use of GNNs in the local step is motivated by their demonstrated alignment with dynamic programming (DP), which can be seen as underlying shortest path computations. Recent works (e.g., Xu et al.,  2019; Dudzik et al., 2022) show that GNNs can approximate DP-like processes leveraging local message-passing. We have revised the manuscript to clarify these connections and provide the references above:
> > > > > >
> > > > > > ``The use of GNNs in the local step is motivated by their demonstrated alignment with dynamic programming (DP). DP underlies many reasoning tasks, including shortest paths which can be solved using the Bellman-Ford algorithm. Recent works have shown that GNNs align well with DP, meaning their computation structures naturally reflect the algorithmic processes of tasks like shortest path computation, which improves learning efficiency and generalization (Xu et al., 2019)[Theorem 3.6]. In (Dudzik et al., 2022), this alignment has been theoretically quantified, suggesting that GNN architectures are particularly well-suited for reasoning tasks where DP plays a central role.''

---

> ### Author Response · Authors · 2024-11-25
> **Part 5**
>
> > **This doesn't answer my question on " it is unclear how the node features contribute meaningfully in this context. Providing examples or intuitive explanations would help clarify this aspect."**
>
> > **The reason for this request is that, unlike many other graph properties, shortest path distances depend solely on graph structures and are independent of node features. Therefore, it is unclear what kind of node features was used in the experiments where GNNs were applied for local steps. Please include a concrete example to clarify this.**
>
> In order to train the GNN, we proceed as follows. We sample a training set of ER graphs of size $n$ and generate random input signals $\mathbf{X} \in \mathbb{R}^{n \times r}$ where each column corresponds to a seed and one-hot encodes which node is a seed for a given graph. The outputs have the same dimensions as the inputs, $\mathbf{Y} \in \mathbb{R}^{n \times r}$, and correspond to the exact shortest path distances between nodes $u \in V$ and seeds $s \in S$, i.e., $[\mathbf{Y}]_{us} = d(u,s)$.
>
> *I.e., the GNN can be thought of as being trained via imitation learning of breadth-first search.*
>
> > **I fail to see how this approach ensures GNN transferability. Unlike other graph properties, distance information does not appear to adhere to consistent patterns, as the same node can have varying distances to different nodes.**
>
> Thanks for this point. We use the same GNN but increase the number of seed sets by running the GNN inference step $\sqrt{N}/\sqrt{n}$ times, where $n$ is the size of the smaller training graph and $N$ is the size of the large target graph. This corresponds to $\sqrt{N}/\sqrt{n}$ different random seed sets, whose embeddings are then appended to form a $\sqrt{N}$-dimensional embedding per node.
>
> The transferability paradigm we consider assumes that the graphs and the inputs are sampled from the same random graph-and-signal model (as in, e.g., Ruiz et al., 2020; Keriven et al., 2020; Levie et al., 2021).
> Transferability works because as the graph size increases, we increase the number of seeds, essentially making sure that the one-hot embeddings $\mathbf{x}_n$ are close to $\mathbf{x}_N$ in $L_2$.
>
> In other words, the induced stepfunctions $X_n(u) = \sum_{i=1}^n [\mathbf{x}_n]_i \mathbb{I}(u \in [(i-1)/n,i/n])$ and respectively $X_m$ are close in $L_2$ norm. Moreover, intuitively increasing the number of seeds as the graph size increases ensures that the distance from nodes to *any seed* does not explode with the graph size.

---

> > ### Comment · Reviewer_FQhS · 2024-11-26
> > **Official Comment by Reviewer FQhS**
> >
> > I would like to sincerely thank the authors for their responses and efforts in improving the paper. However, I still have some questions/comments:
> >
> > > "However, our motivation stems from scenarios where such indexes are impractical---for example, in dynamic networks where distances need to be frequently updated, or in applications ..."
> >
> > In dynamic networks, the index sizes for shortest path distances are typically not significantly larger than those in static networks. Recent works, such as [1, 2], demonstrate efficient approaches to maintain shortest path distance indexes in dynamic settings.
> >
> > > "... Recent works have shown that GNNs align well with DP, meaning their computation structures naturally reflect the algorithmic processes of tasks like shortest path computation, which improves learning efficiency and generalization (Xu et al., 2019)[Theorem 3.6]. In (Dudzik et al., 2022), this alignment has been theoretically quantified, suggesting that GNN architectures are particularly well-suited for reasoning tasks where DP plays a central role.''
> >
> > Theorem 3.6 in Xu et al., 2019 addresses algorithmic alignment and sample complexity. It is based on the assumption that certain neural networks can simulate specific functions. I can see how the GNN architecture discussed in (Xu et al., 2019) can simulate the shortest path computation. However, this does not directly imply that the GNN proposed in this work can do so, especially given that the architecture discussed in (Xu et al., 2019) differs from the one proposed here. For example, they simulate the Bellman-Ford algorithm with MLP; however, the proposed GNN architecture in this work lack an MLP. The effectiveness of node features is unclear too.
> >
> > On the other hand, if the GNN design in this work primarily applies existing techniques, it raises questions about its novelty and contributions from an ML perspective (as research on approximation algorithms would typically align with venues like SODA).
> >
> > Thus, I would encourage the authors to present formal statements and illustrative examples, similar to those in [3] (the paper was also suggested by Reviewer oaDP), to clearly articulate the novelty of the proposed GNN architecture and provide a theoretical proof to support the claims.
> >
> > [1] Relative Subboundedness of Contraction Hierarchy and Hierarchical 2-Hop Index in Dynamic Road Networks, Zhang, et al., SIGMOD 2022
> >
> > [2] BatchHL+: Batch Dynamic Labelling for Distance Queries on Large-Scale Networks, Farhan, et al., The VLDB Journal 2023
> >
> > [3] Shouheng et al., Local Vertex Colouring Graph Neural Networks, ICML 2023.

---

> > > ### Author Response · Authors · 2024-11-26
> > > **Part 6**
> > >
> > > > **Theorem 3.6 in Xu et al., 2019 addresses algorithmic alignment and sample complexity. It is based on the assumption that certain neural networks can simulate specific functions. I can see how the GNN architecture discussed in (Xu et al., 2019) can simulate the shortest path computation. However, this does not directly imply that the GNN proposed in this work can do so, especially given that the architecture discussed in (Xu et al., 2019) differs from the one proposed here. For example, they simulate the Bellman-Ford algorithm with MLP; however, the proposed GNN architecture in this work lack an MLP. The effectiveness of node features is unclear too.**
> > >
> > > Thank you for your comments. To clarify, the GNN in (Xu et al., 2019) uses an MLP on the feature dimension of the data, while the node-level operation is message-passing/neighborhood aggregation. This distinction is consistent with the design of the GIN model with max aggregation, which is explicitly described in equation (2.2) of (Xu et al., 2019) and is one of the architectures we employ in our local step. Therefore, their theoretical result does directly apply to our setup, implying that, with sufficient samples, the GIN architecture in our local step is capable of simulating the Bellman-Ford algorithm.
> > >
> > > We believe this connection is clear and aligns with the theoretical framework established in (Xu et al., 2019)
> > > > **On the other hand, if the GNN design in this work primarily applies existing techniques, it raises questions about its novelty and contributions from an ML perspective (as research on approximation algorithms would typically align with venues like SODA).**
> > >
> > > Our algorithm builds upon the combination of the local-global framework of Bourgain/Das Sarma and GNNs. While individually these components are not novel, the integration of GNNs into the local-global framework provides a fresh perspective. This approach is novel in its own right (note that our algorithm differs from (Awasthi et al., 2022) in that while they use several GNNs, we only employ one).
> > >
> > > That said, the primary goal of our work is not the design of a new ML model but rather the average-case analysis of local-global algorithms, which motivated the replacement of the BFS step with GNNs. This substitution allowed us to bridge theoretical insights with practical considerations and empirically validate the alignment of GNNs with the local-global methodology.

---

### Official Review · Reviewer_oaDP · 2024-11-01

**Soundness:** 3
**Presentation:** 3
**Contribution:** 3
**Rating:** 6
**Confidence:** 4

**Summary:**

This paper explores the limitations of local message passing in GNNs for solving shortest path problems in graphs. Building on prior research by Awasthi et al. (2022), the authors conduct a theoretical analysis of local-global algorithms applied to Erdős-Rényi (ER) random graphs, demonstrating that these algorithms can achieve improved accuracy in estimating shortest distances with reduced distortion and embedding dimensions. They propose an enhancement that integrates GNNs into the local computation phase to boost efficiency. Empirical results confirm that the GNN-augmented algorithms significantly outperform traditional methods on both ER graphs and benchmark datasets, highlighting their applicability in real-world scenarios, such as social networks. Overall, this work advances the understanding of effective strategies for shortest path computation in complex networks.

**Strengths:**

1. The paper presents a novel integration of GNNs into existing local-global algorithms for shortest path computation. This combination of traditional algorithms with modern neural approaches is timely and relevant, particularly as GNNs gain traction in graph-related problems.

2. The methodology is well-defined, with a clear explanation of the algorithmic framework. The theoretical analysis provided adds rigor to the claims, showing a strong foundation for the proposed enhancements.

3. The writing is generally clear and well-structured, allowing readers to follow the development of ideas easily.

**Weaknesses:**

1. While the authors claim superior performance, there is limited comparative analysis with other shortest-path based GNN approaches [1, 2, 3]. A more detailed benchmarking against established methods could validate the claims and provide a clearer context for the contributions.

2. The theoretical analysis is strong but could be expanded to cover edge cases or graph types beyond Erdős-Rényi. This would enhance the generalizability of the findings and provide deeper insights into the algorithm’s performance.

3. More detailed information / empirical study on the computational complexity and memory requirements of the proposed method would be beneficial.

References:

[1] Shouheng et al., Local Vertex Colouring Graph Neural Networks, ICML 2023.

[2] Bohang et al., Rethinking the Expressive Power of GNNs via Graph Biconnectivity, ICLR 2023.

[3] Petar et al., Neural Execution of Graph Algorithms, ICLR 2020.

**Questions:**

1. Could you elaborate on the specific architectural choices made for the GNNs? How do these choices influence their performance when transferred to larger graphs?

2. What criteria were used to evaluate the transferability of GNNs across varying graph sizes? Were there instances where transferability was not achieved?

3. Could you elaborate on how your GNN-enhanced approach stacks up against other leading algorithms in terms of efficiency and accuracy? A detailed comparison would help clarify the unique contributions of your work.

4. How do your theoretical bounds perform under various graph distributions beyond the Erdős-Rényi model?

5. In the experiments section, authors mention a notable performance increase when the GNN is trained on graphs with 200 nodes for the larger $n^{′}$-node graph. What are the practical implications of this finding, especially when graph sizes vary widely?

---

> ### Author Response · Authors · 2024-11-20
> **Thank you for your review. Your comments are addressed point-by-point below.**
>
> >**While the authors claim superior performance, there is limited comparative analysis with other shortest-path based GNN approaches [1, 2, 3]. A more detailed benchmarking against established methods could validate the claims and provide a clearer context for the contributions.**
>
> >**Refs.:**
>
> >**[1] Shouheng et al., Local Vertex Colouring Graph Neural Networks, ICML 2023.**
>
> >**[2] Bohang et al., Rethinking the Expressive Power of GNNs via Graph Biconnectivity, ICLR 2023.**
>
> >**[3] Petar et al., Neural Execution of Graph Algorithms, ICLR 2020.**
>
> Thank you for raising this concern and for pointing us to these references. These are very relevant to our paper and will be cited in the revision.
>
> Regarding performance, we respectfully point out that we do not claim superior performance. The main contribution of our manuscript is an analysis of local-global shortest path (SP) algorithms on ER graphs.
> The idea was to initiate the study of average case analysis in the same spirit as the classical works in the CS literature for NP hard problems.
> Our goal is slightly different though, since the underlying problem is not NP-hard, and the average case analysis is used for understanding possible reduction in complexity. We will add a few lines along this direction in the introduction to improve clarity in our stated motivation.
>
> Since our theorems show that a lower embedding size is achievable on ER graphs, we were motivated to explore the possibility of replacing the breadth-first search (BFS) in the local step---which does not have a fixed width and depth for arbitrary graphs---with a GNN---which has fixed width and depth---as an empirical contribution to reduce computational complexity. The reasoning for using a GNN in the local step of such algorithms, as opposed to using GNNs for end-to-end SP computations, has to do with results by Loukas (2020) which show that it is impossible for GNNs with fixed depth and width to learn SPs on arbitrary graphs. This is a key difference between our paper and [1]--[3], which use GNNs to do end-to-end shortest path computations, so we do not believe an empirical comparison applies. We note however that it would be possible to incorporate any (local) GNN architecture, including those in [1]--[3], in the local step of Algorithm 1. We leave this as an interesting direction for future work. Thanks once again for the suggestion.
>
> >**The theoretical analysis is strong but could be expanded to cover edge cases or graph types beyond Erdős-Rényi. This would enhance the generalizability of the findings and provide deeper insights into the algorithm’s performance.**
>
> This is a good point, and we acknowledge the limitations of focusing solely on ER graphs. However, our choice was deliberate: starting with simpler models like ER graphs allows us to develop the theoretical tools and insights needed for analyzing algorithms on more complex graph structures. Even for ER graphs, our analysis is nontrivial, as it depends on results about the growth of local neighborhoods.
>
> We agree that extending this analysis to more realistic graph models is an important next step. Specifically, we are exploring inhomogeneous random graphs, which better model social networks, and expect our local neighborhood expansion results to hold in this setting. Thank you for raising this concern.
>
> >**More detailed information / empirical study on the computational complexity and memory requirements of the proposed method would be beneficial.**
>
> Thank you for bringing this up. We refer the reviewer to Figure 3c, which shows the time required to generate the distances between the seeds and all nodes (i.e., the sketches) for both the BFS- and the GNN-based approach.
>
> >**Could you elaborate on the specific architectural choices made for the GNNs? How do these choices influence their performance when transferred to larger graphs? What criteria were used to evaluate the transferability of GNNs across varying graph sizes? Were there instances where transferability was not achieved**
>
> For each of the eight training graph sizes, we evaluated four GNN architectures (GCN, GraphSAGE, GAT, and GIN) across nine combinations of width and depth, as detailed in the experimental section and appendices. Using MSE to evaluate performance, we selected the GNN with the lowest MSE for node embedding calculations in the local step.
>
> Selecting the best-performing GNN ensured high-quality node embeddings, improving shortest-path approximations in the local-global framework, as shown in Experiment 3. GNNs trained on larger graphs generally enhanced transferability by capturing more relevant structural patterns.
>
> Transferability was not achieved when the GNN in the local step was poorly trained or lacked sufficient depth to encode meaningful embeddings. While model tuning helped identify suitable architectures, understanding the theoretical relationship between GNN design and embedding quality remains an important direction for future research.

---

> > ### Author Response · Authors · 2024-11-20
> > **Part 2**
> >
> > >**Could you elaborate on how your GNN-enhanced approach stacks up against other leading algorithms in terms of efficiency and accuracy? A detailed comparison would help clarify the unique contributions of your work.**
> >
> > We thank the reviewer for this feedback. As noted in our response to your first question, we will include a comparison of our theoretical results with Theorem 1 in Awasthi et al. (2022) in the paper, noting that we achieve a better approximation factor and improved depth-width trade-off (i.e., embedding size) on Erdős–Rényi graphs.
> >
> > Regarding empirical comparisons, we respectfully point out the primary goal of our paper was not to introduce a new state-of-the-art method for SP calculations, but rather to provide a theoretical analysis of existing local-global SP algorithms on ER graphs.
> > The idea behind this analysis is to establish guarantees for the approximation factor/embedding size of these algorithms that are more akin to average-case guarantees commonly studied in theoretical computer science, as existing theoretical analyses of local-global SP algorithms have only focused on worst-case guarantees.
> >
> > Since our theorems show that a lower embedding size is achievable on ER graphs, we were motivated to explore the possibility of replacing the breadth-first search (BFS) in the local step---which does not have a fixed width and depth for arbitrary graphs---with a GNN---which has fixed width and depth---as an empirical contribution to reduce computational complexity. Another reason to use an ML model in the local step is that, once the ML model is trained, the inference step on a new seed set, or even on a new graph, is fast and cheap, and can even be done online, thus freeing some memory. This is not the case for BFS, which requires exact shortest path computations for the new seed set and/or graph.
> >
> > >**How do your theoretical bounds perform under various graph distributions beyond the Erdős-Rényi model?**
> >
> > Our analysis for ER graphs depends on results about the growth of local neighborhoods. As such, we expect the methodology developed here to extend to other graphs with local expansion properties, such as inhomogeneous random graphs and expanders. A specific type of inhomogeneous random graph for which we expect the same relationship between approximation factor and embedding size are stochastic block models; these quantities would only differ from the same quantities for ER graphs by a constant factor.
> >
> > >**In the experiments section, authors mention a notable performance increase when the GNN is trained on graphs with 200 nodes for the larger $n'$-node graph. What are the practical implications of this finding, especially when graph sizes vary widely?**
> >
> > Experiment 3 shows that GNNs trained on ER graphs with as few as 200 nodes can produce node embeddings with competitive performance on test graphs with 12,800 nodes. While training on larger graphs improves results, this finding suggests that reasonable performance can still be achieved with smaller training graphs, which is especially useful in resource-constrained settings.

---

### Official Review · Reviewer_6QDQ · 2024-11-04

**Soundness:** 3
**Presentation:** 2
**Contribution:** 3
**Rating:** 8
**Confidence:** 4

**Summary:**

This paper analyzes hybrid algorithms that combine local and global approaches to solve shortest path problems on graphs, especially focusing on Erdős-Rényi random graphs. By adding Graph Neural Networks (GNNs) into the local phase, the algorithm becomes faster and more scalable for big graph networks. Through theoretical analysis on Erdős-Rényi random graphs, the authors show that their method provides tight distance estimates for most node pairs. Experiments on both synthetic and real-world graphs highlight the improved performance of the GNN-enhanced approach.

**Strengths:**

The paper introduces an innovative hybrid algorithm for shortest path approximation on random graphs that uses the locality of GNNs and the global approximation bounds provided by Bourgain's theorem.

**Weaknesses:**

Although inspired by Awasthi et al.'s framework, there isn’t a direct comparison of results, such as accuracy or runtime metrics. The proof sections are a bit difficult to follow, especially for people without a sound background in this field. A detailed table introducing all the variables introduced in the algorithms and proofs would help to understand them easily.

**Questions:**

1. How does this method perform on large-scale networks with millions of nodes, like social or biological networks? SNAP dataset collection of Stanford University has some very large social network datasets.
2. How does the GNN-based approach handle real-world sparse networks where long-range dependencies are more challenging? Could alternative GNN architectures improve performance in such cases?
3. Could the approach be adapted to handle other graph structures (like protein graphs) effectively?

**Details Of Ethics Concerns:**

In line 830, a GitHub link to the project code is mentioned, where it can be found that this is a work from ruiz-lab and from the Applied Math and Statistics department of John Hopkins.

---

> ### Author Response · Authors · 2024-11-20
> **Thank you for your review. Your comments are addressed point-by-point below.**
>
> >**Although inspired by Awasthi et al.'s framework, there isn’t a direct comparison of results, such as accuracy or runtime metrics.**
>
> We thank the reviewer for this feedback. We will include the following comparison of our theoretical results with Theorem 1 in Awasthi et al.:
>
> ``In the worst case analysis, Sarma et al. (2010), Matousek (1996), and Awasthi et al. (2022) showed that a $(2c-1)$-factor upper bound for all pairs and a $\frac{1}{2c-1}$-factor lower bound of shortest paths can be achieved using local-global algorithms with an embedding dimension of $\Omega(n^{1/c} \log n)$ for $c>1$.
> Our results require an embedding dimension of $\Omega (n^{3-2c-\log 2}\log{n})$ for a $(2c-1)$-factor upper bound and $\Omega (n^{1/(2c-1)}\log{n})$ for a $\frac{1}{2c-1}$-factor lower bound with high probability for most pairs of nodes. Thus, in random graphs, we achieve an improvement in the embedding dimension.
> On the downside, the results for random graphs do not provide approximate distances for all pairs of nodes. We leave this as an interesting direction for future research.''
>
> Regarding empirical comparisons, we note that the code for GNN+ is not publicly available, which limits our ability to conduct direct experimental evaluations. Additionally, Awasthi et al.'s GNN+ framework comprises multiple GNNs, whereas the GNN-augmented algorithm we consider relies on a single GNN, so these approaches are somewhat different in their design and scope.
>
> >**The proof sections are a bit difficult to follow, especially for people without a sound background in this field. A detailed table introducing all the variables introduced in the algorithms and proofs would help to understand them easily.**
>
> Thank you for the feedback. To improve clarity, we have revised the theorem statements and proofs for the lower and upper bound distortions in Section 3. Additionally, we now redefine all relevant variables at the beginning of each section to make the notation more accessible and easier to follow.
>
> >**How does this method perform on large-scale networks with millions of nodes, like social or biological networks? SNAP dataset collection of Stanford University has some very large social network datasets.**
>
> Thank you for the suggestion. We acknowledge that our experiments so far have been limited to smaller networks. While trying to work with batches of 100 graphs of size 12,800 nodes, we had kernel crashes, which prevented us from scaling up further at the time. However, as suggested by the reviewer, we are currently repeating Experiment 3 on a cloud service using SNAP's Amazon co-purchasing network, with around 100k nodes, and Youtube network, with around 1 million nodes, as the test graph. These results will be included in Section 4.3 and reported as they become available.
>
> >**How does the GNN-based approach handle real-world sparse networks where long-range dependencies are more challenging? Could alternative GNN architectures improve performance in such cases?**
>
> Our method is based on local neighborhood expansions, so it does not capture long-range dependencies. While such dependencies are less common in the sparse Erdős-Rényi (ER) graphs analyzed in our paper, we acknowledge this as a limitation and will highlight it in the revised version. That said, our results already hold for sparse ER graphs, which are precisely the type of graphs we focus on in this work. Furthermore, we expect the methodology developed here to extend to other graphs with local expansion properties, such as inhomogeneous random graphs and expanders. We are actively exploring these extensions.
>
> In our experiments, we demonstrated that GNNs struggle to predict distances longer than their depth. But in the local-global framework, we hypothesize that this limitation, i.e., the existence of seed-node pairs for which the GNN distances/sketches saturate, can be advantageous in the context of Bourgain's algorithm (lower bound), as it effectively ``prunes'' poor-quality seed nodes that are far from most other nodes. This will happen when nodes $u,v$ have almost the same distance to the seed $s$, improving the lower-bound computation $\max_{s,s' \in S} |d(u,s)-d(s',v)|$.
>
> We leave the theoretical exploration of the relationship between the performance of GNN-enhanced local-global algorithms and the GNN architecture an avenue for future research.

---

> > ### Author Response · Authors · 2024-11-20
> > **Part 2**
> >
> > >**Could the approach be adapted to handle other graph structures (like protein graphs) effectively?**
> >
> > Thank you for the interesting question. Our method was developed and tested specifically on ER graphs, and the GNNs used in our approach have been trained only on this type of graph. While we expect the methodology developed here to extend to other graphs with local expansion properties, such as inhomogeneous random graphs and expanders, and while Experiment 3 demonstrates some transferability to real-world undirected, unweighted graphs, we do not yet know how well our method would perform on protein graphs or how it might need to be adapted for such structures.

---

### Author Response · Authors · 2024-11-20
**Rebuttal summary**

We’d like to thank the area chair and reviewers for their thoughtful feedback and for managing the review process for our paper.

We appreciate the reviewers highlighting the strengths of our work, such as the *"innovative hybrid algorithm" (Reviewer 6QDQ)* combining GNNs with Bourgain’s theorem, the *"timely and relevant" (Reviewer oaDP)* integration of GNNs into local-global frameworks, and the *"rigor" (Reviewer oaDP)* in the theoretical analysis demonstrating tight distance estimates for most node pairs.

We would also like to clarify that **our work is fundamentally about the theoretical analysis of local-global shortest-path algorithms on Erdős-Rényi (ER) graphs**. Here is what we focused on in the revision:

### **Emphasis on Theory**
- Highlighted that our main contribution is the theoretical guarantees for approximation factors and embedding sizes on ER graphs, focusing on average-case analysis rather than empirical performance.
- Included comparisons with the theoretical results from Awasthi et al. (2022), emphasizing the lower embedding dimension requirements for ER random graphs.

### **Experiments as Illustrations**
- Clarified that our experiments are intended to illustrate our theoretical findings as opposed to competing with state-of-the-art methods. We are extending these experiments to larger datasets (Amazon co-purchasing and Youtube SNAP networks) to address scalability questions.

### **Acknowledging Scope and Extensions**
- Addressed limitations in scalability and transferability, framing this work as theoretical groundwork and methodology building for future extensions to more complex graph models like inhomogeneous random graphs.

### **Sharper Framing**
- Revised the motivation and related work sections to better reflect the theoretical focus and added missing citations suggested by the reviewers.

---

> ### Author Response · Authors · 2024-11-26
> **Update: experiments on larger graphs added to Figures 4(d)--4(f)**
>
> Thanks again to the AC and the reviewers for their efforts in reviewing our paper. As an update, additional experiments on larger graphs with up to 100k nodes are now reported in Figures 4(d)--4(f) of the manuscript. We could not scale to larger graphs due to limited computational resources. Given the primarily theoretical nature of our paper, we hope the reviewers can understand.

---

### Comment · Area_Chair_M9vv · 2024-11-23

Dear Reviewers,

The authors have uploaded their rebuttal. Please take this opportunity to discuss any concerns you may have with the authors.

AC

---

### Meta-Review · Area_Chair_M9vv · 2024-12-20

**Metareview:**

The paper introduces a hybrid algorithm that combines local-global approaches with Graph Neural Networks (GNNs) to solve the shortest path problem on graphs, focusing on Erdős-Rényi (ER) random graphs. The GNNs are used in the local phase to improve computational efficiency and scalability, particularly for large networks. The paper presents theoretical analysis showing tight distance estimates for most node pairs and empirical results demonstrating improved performance on both synthetic and real-world graphs.

While the proposed model shows promising results, there are several weaknesses that still need to be addressed:

1. The transferability experiment is somewhat limited in scope. The authors use the same lambda value for both the training and testing graphs, but it is crucial to conduct experiments where the lambda values for the training and test graphs differ. If the transferability of the proposed method holds only for graph families with identical lambda values, its applicability would be quite restricted. Therefore, it is important to include experiments where GNNs trained on graphs with lambda = 4 are applied to graphs with lambda = 5 to better assess generalizability.
2. Some results from the transferability experiment are not fully supported by the theory proposed by the authors. For instance, in Figures 4(d) and 4(f), there is a sudden increase in MSE as the number of nodes (n) grows. The authors have not provided an explanation for this phenomenon, either through theoretical analysis or by offering hypotheses to explain the observed behavior.

Based on these weaknesses, we recommend rejecting this paper. We hope this feedback helps the authors improve their paper.

**Additional Comments On Reviewer Discussion:**

In their rebuttal, the authors made several improvements, including clarifications and updates to the presentation, which help reviewers better understand the contributions of the paper. However, the reviewers’ concerns regarding the transferability remain unresolved. As a result, I recommend rejection based on the reviewers’ feedback.

---

### Decision · Program_Chairs · 2025-01-22

Reject